# GRAPHON BASED CLUSTERING AND TESTING OF NETWORKS: ALGORITHMS AND THEORY

**Mahalakshmi Sabanayagam**
Technical University of Munich
`maha.sabanayagam@tum.de`

**Leena C. Vankadara**
University of Tübingen
`leena.chennuru-vankadara@uni-tuebingen.de`

**Debarghya Ghoshdastidar**
Technical University of Munich
Munich Data Science Institute
`ghoshdas@in.tum.de`

## ABSTRACT

Network-valued data are encountered in a wide range of applications, and pose challenges in learning due to their complex structure and absence of vertex correspondence. Typical examples of such problems include classification or grouping of protein structures and social networks. Various methods, ranging from graph kernels to graph neural networks, have been proposed that achieve some success in graph classification problems. However, most methods have limited theoretical justification, and their applicability beyond classification remains unexplored. In this work, we propose methods for clustering multiple graphs, without vertex correspondence, that are inspired by the recent literature on estimating graphons—symmetric functions corresponding to infinite vertex limit of graphs. We propose a novel graph distance based on sorting-and-smoothing graphon estimators. Using the proposed graph distance, we present two clustering algorithms and show that they achieve state-of-the-art results. We prove the statistical consistency of both algorithms under Lipschitz assumptions on the graph degrees. We further study the applicability of the proposed distance for graph two-sample testing problems.

## 1 INTRODUCTION

Machine learning on graphs has evolved considerably over the past two decades. The traditional view towards network analysis is limited to modelling interactions among entities of interest, for instance social networks or world wide web, and learning algorithms based on graph theory have been commonly used to solve these problems (Von Luxburg, 2007; Yan et al., 2006). However, recent applications in bioinformatics and other disciplines require a different perspective, where the networks are the quantities of interest. For instance, it is of practical interest to classify protein structures as enzyme or non-enzyme (Dobson & Doig, 2003) or detect topological changes in brain networks caused by Alzheimer's disease (Stam et al., 2007). In this paper, learning from network-valued data refers to clustering where each network is treated as an entity, as opposed to the traditional network analysis problems that involve a single network of interactions (Newman, 2003).

Machine learning on network-valued data has been an active area of research in recent years, although most works focus on the network classification problem. The generic approach is to convert the network-valued data into a standard representation. Graph neural networks are commonly used for network embedding, that is, finding Euclidean representations of each network that can be further used in standard machine learning models (Narayanan et al., 2017; Xu et al., 2019). In contrast, graph kernels capture similarities between pairs of networks that can be used in kernel based learning algorithms (Shervashidze et al., 2011; Kondor & Pan, 2016; Togninalli et al., 2019). In particular, the graph neural tangent kernel defines a graph kernel that corresponds to infinitely wide graph neural networks, and typically outperforms neural networks in classification tasks (Du et al., 2019). A more classical equivalent for graph kernels is to define metrics that characterise the distances be-

tween pairs of graphs (Bunke & Shearer, 1998), but there has been limited research on designing efficient graph distances and developing algorithms for clustering network-valued data.

The motivation for this paper stems from two shortcomings in the literature on network-valued data analysis: first, the efficacy of existing kernels or embeddings have not been studied beyond network classification, and second is the lack of theoretical analysis of these methods, particularly in the small sample setting. Generalisation error bounds for graph kernel based learning exist (Du et al., 2019), but these bounds, based on learning theory, are meaningful only when many networks are available. However, in many applications, one needs to learn from a small population of large networks and, in such cases, an informative statistical analysis should consider the small sample, large graph regime. To address this issue, we take inspiration from the recent statistics literature on graph two-sample testing—given two (populations of) large graphs, the goal is to decide if they are from same statistical model or not. Although most theoretical studies in graph two-sample testing focus on graph with vertex correspondence (Tang et al., 2017a; Ghoshdastidar & von Luxburg, 2018), some works address the problem of testing graphs on different vertex sets either by defining distances between graphs (Tang et al., 2017b; Agterberg et al., 2020) or by representing networks in terms of pre-specified network statistics (Ghoshdastidar et al., 2017). The use of network statistics for clustering network-valued data is studied in Mukherjee et al. (2017). Another fundamental approach for dealing with graphs of different sizes is graph matching, where the objective is to determine the vertex correspondence. Graph matching is often solved by formulating it as an optimization problem (Zaslavskiy et al., 2008; Guo et al., 2019) or defining graph edit distance between the graphs (Riesen & Bunke, 2009; Gao et al., 2010). Although there is extensive research on graph matching, the efficacy of these methods in clustering network-valued data remains unexplored.

**Contribution and organisation.** In this work, we follow the approach of defining meaningful graph distances based on statistical models, and use the proposed graph distance in the context of learning from networks without vertex correspondence. In particular, we propose graph distances based on graphons. Graphons are symmetric bivariate functions that represent the limiting structure for a sequence of graphs with increasing number of nodes (Lovász & Szegedy, 2006), but can be also viewed as a nonparametric statistical model for exchangeable random graphs (Diaconis & Janson, 2007; Bickel & Chen, 2009). The latter perspective is useful for the purpose of machine learning since it allows us to view the multiple graphs as random samples drawn from one or more graphon models. This perspective forms the basis of our contributions, which are listed below:

**1)** In Section 2, we propose a distance between two networks, that do not have vertex correspondence and could have different number of vertices. We view the networks as random samples from (unknown) graphons, and propose a graph distance that estimates the $L_2$-distance between the graphons. The distance is inspired by the sorting-and-smoothing graphon estimator (Chan & Airoldi, 2014).

**2)** In Section 3, we present two algorithms for clustering network-valued data based on the proposed graph distance: a distance-based spectral clustering algorithm, and a similarity based semi-definite programming (SDP) approach. We derive performance guarantees for both algorithms under the assumption that the networks are sampled from graphons satisfying certain smoothness conditions.

**3)** We empirically compare the performance of our algorithms with other clustering strategies based on graph kernels, graph matching, network statistics etc. and show that, on both simulated and real data, our graph distance-based spectral clustering algorithm outperforms others while the SDP approach also shows reasonable performance, and they also scale to large networks (Section 3.3).

**4)** Inspired by the success of the proposed graph distance in clustering, we use the distance for graph two-sample testing. In Section 4, we show that the proposed two-sample test is statistically consistent for large graphs, and also demonstrate the efficacy of the test through numerical simulation.

We provide further discussion in Section 5 and present the proofs of theoretical results in Appendix.

## 2 GRAPH DISTANCE BASED ON GRAPHONS

Clustering or testing of multiple networks requires a notion of distance between the networks. In this section, we present a transformation that converts graphs of different sizes into a fixed size representation, and subsequently, propose a graph distance inspired by the theory of graphons. We first provide some background on graphons and graphon estimation. Graphon has been studied in

the literature from two perspectives: as limiting structure for infinite sequence of growing graphs (Lovász & Szegedy, 2006), or as exchangeable random graph model. In this paper, we follow the latter perspective. A random graph is said to be exchangeable if its distribution is invariant under permutation of nodes. Diaconis & Janson (2007) showed that any statistical model that generates exchangeable random graphs can be characterised by graphons, as introduced by Lovász & Szegedy (2006). Formally, a graphon is a symmetric measurable continuous function $w : [0,1]^2 \to [0,1]$ where $w(x, y)$ can be interpreted as the link probability between two nodes of the graph that are assigned values $x$ and $y$, respectively. This interpretation propounds the following two stage sampling procedure for graphons. To sample a random graph $G$ with $n$ nodes from a graphon $w$, in the first stage, one samples $n$ variables $U_1, \ldots, U_n$ uniformly from $[0,1]$ and constructs a latent mapping between the sampled points and the node labels. In the second stage, edges between any two nodes $i, j$ are randomly added based on the link probability $w(U_i, U_j)$. Mathematically, if we abuse notation to denote the adjacency matrix by $G \in \{0,1\}^{n \times n}$, we have

$$U_1, \ldots, U_n \overset{\text{iid}}{\sim} Uniform[0,1] \qquad \text{and} \qquad G_{ij}|U_i, U_j \sim Bernoulli(w(U_i, U_j)) \text{ for all } i < j.$$

We consider problems involving multiple networks sampled independently from the same (or different) graphons. We make the following smoothness assumptions on the graphons.

**Assumption 1 (Lipschitz continuous)** *A graphon $w$ is Lipschitz continuous with constant $L$ if*

$$|w(u,v) - w(u',v')| \leq L\sqrt{(u-u')^2 + (v-v')^2} \qquad \text{for every } u, v, u', v' \in [0,1].$$

**Assumption 2 (Two-sided Lipschitz degree)** *A graphon $w$ has two-sided Lipschitz degree with constants $\lambda_1, \lambda_2 > 0$ if its expected degree function $g$, defined by $g(u) = \int_0^1 w(u,v)\mathrm{d}v$, satisfies*

$$\lambda_2|u - u'| \leq |g(u) - g(u')| \leq \lambda_1|u - u'| \qquad \text{for every } u, u' \in [0,1].$$

One of the challenges in graphon estimation is due to the issue of non-identifiability, that is, different graphon functions $w$ can generate the same random graph model. In particular, two graphons $w$ and $w'$ generate the same random graph model if they are weakly isomorphic—there exist two measure preserving transformations $\phi, \phi' : [0,1] \to [0,1]$ such that $w(\phi(u), \phi(v)) = w'(\phi'(u), \phi'(v))$. Moreover, the converse also holds meaning that such transformations are known to be the only source of non-identifiability (Diaconis & Janson, 2007). This weak isomorphism induces equivalence classes on the space of graphons. Since our goal is only to cluster graphs belonging to random graph models, we simply make the following assumption on our graphons.

**Assumption 3 (Equivalence classes)** *Any reference to $K$ graphons, $w_1, \ldots, w_K$, assumes that, for every $i, j$, either $w_i = w_j$ or $w_i$ and $w_j$ belong to different equivalence classes. Furthermore, without loss of generality, we assume that every graphon $w_i$ is represented such that the corresponding degree function $g_i$ is non-decreasing.*

**Remark on the necessity of Assumptions 1–3.** Assumption 1 is standard in graphon estimation literature (Klopp et al., 2017) since it avoids graphons corresponding to inhomogeneous random graph models. It is known that two graphs from widely separated inhomogeneous models (in $L_2$-distance) are statistically indistinguishable (Ghoshdastidar et al., 2020), and hence, it is essential to ignore such models to derive meaningful guarantees. Assumption 2 ensures that, under a measure-preserving transformation, the graphon has strictly increasing degree function, which is a canonical representation of an equivalence class of graphons (Bickel & Chen, 2009). Assumption 3 is needed since graphons can only be estimated up to measure-preserving transformation. As noted above, it is inconsequential for all practical purposes but simplifies the theoretical exposition.

**Graph transformation.** In order to deal with multiple graphs and measure distances among pairs of graphs, we require a transformation that maps all graphs into a common metric space—the space of all $n_0 \times n_0$ symmetric matrices for some integer $n_0$. While the graphon estimation literature provides several consistent estimators (Klopp et al., 2017; Zhang et al., 2017), only the histogram based sorting-and-smoothing graphon estimator of Chan & Airoldi (2014) can be adapted to meet the above requirement. We use the following graph transformation, inspired by Chan & Airoldi (2014). The adjacency matrix $G$ of size $n \times n$ is first reordered based on a non-unique permutation $\sigma$, such that the empirical degree based on this permutation is monotonically increasing. The degree

sorted adjacency matrix is denoted by $G^\sigma$. It is then transformed to a 'histogram' $A \in \mathbb{R}^{n_0 \times n_0}$ as

$$A_{ij} = \frac{1}{h^2} \sum_{i_1=1}^{h} \sum_{j_1=1}^{h} G^\sigma_{(i-1)h+i_1,(j-1)h+j_1}, \text{ where } h = \left\lfloor \frac{n}{n_0} \right\rfloor \text{ and } \lfloor \cdot \rfloor \text{ is the floor function.} \quad (1)$$

**Proposed graph distance.** Given two directed or undirected graphs $G_1$ and $G_2$ with $n_1$ and $n_2$ nodes, respectively, we apply the transformation (1) to both the graphs with $n_0 \leq \min\{n_1, n_2\}$. We propose to use the graph distance

$$d(G_1, G_2) = \frac{1}{n_0} \|A_1 - A_2\|_F, \quad (2)$$

where $A_1$ and $A_2$ denote the transformed matrices and $\|\cdot\|_F$ denotes the matrix Frobenius norm. Proposition 1 shows that, if $G_1$ and $G_2$ are sampled from two graphons, then the graph distance (2) consistently estimates the $L_2$-distance between the two graphons, which is defined as $\|w_1 - w_2\|_{L_2}^2 = \int_0^1 \int_0^1 (w_1(x, y) - w_2(x, y))^2 \, dx \, dy$.

**Proposition 1 (Graph distance is consistent)** *Let $w_1$ and $w_2$ satisfy Assumptions 1–3. Let $G_1$ and $G_2$ be random graphs with at least $n$ nodes sampled from the graphons $w_1$ and $w_2$, respectively. If $n \to \infty$ and $n_0$ is chosen such that $\frac{n_0^2 \log n}{n} \to 0$, then with high probability (w.h.p.),*

$$\left| \| w_1 - w_2 \|_{L_2} - d(G_1, G_2) \right| = \mathcal{O}\left(\frac{1}{n_0}\right). \quad (3)$$

*Proof sketch.* We define a novel technique for approximating the graphon. The proof in Appendix A.1 first establishes that the approximation error is bounded using Assumption 1. Consequently, a relation between approximated graphons and transformed graphs is derived using lemmas from Chan & Airoldi (2014). Proposition 1 is subsequently proved using the above two results. $\square$

**Remark on Proposition 1 for sparse graphs.** The defined sampling procedure for graphon generates dense graphs, deviating from the real world sparse networks. To adapt it to sparse graphs, one may modify the sampling procedure to $G_{ij}|U_i, U_j \sim Bernoulli(\rho\, w(U_i, U_j))$ where $\rho$ depends only on $n$ (Olhede & Wolfe, 2014). Under this process, the consistency result in Proposition 1 remains unchanged for $\rho = \Omega(\sqrt{\log n/n})$ (proof in Appendix A.2). This bound cannot be improved to the expected real world measure where $\rho = \Omega(\log n/n)$, because of the degree sorting step in (1). Nevertheless, our analysis allows for relatively sparser graphs with strong consistency result.

**Notation.** For ease of exposition, Proposition 1 as well as main results are stated asymptotically using the standard $\mathcal{O}(\cdot)$ and $\Omega(\cdot)$ notations, which subsume absolute and Lipschitz constants. We use "with high probability" (w.h.p.) to state that the probability of an event converges to 1 as $n \to \infty$.

## 3 GRAPH CLUSTERING

We now present the first application of the proposed graph distance (2) in the context of clustering network-valued data. We are particularly interested in the setting where one needs to cluster a small population of large graphs, that is, minimum graph size $n$ grows faster than the sample size $m$. This scenario is relevant in practice as bioinformatics or neuroscience application often deals with very few graphs (see real datasets in Section 3.3). Theoretically, this perspective complements guarantees for (graph) kernels that are applicable only in supervised setting and large sample regime, $m \to \infty$. In contrast, our guarantees are more conclusive for bounded $m$ and large graph size, $n \to \infty$.

### 3.1 DISTANCE BASED SPECTRAL CLUSTERING (DSC)

Given $m$ graphs with adjacency matrices $G_1, ..., G_m$, we propose a distance based clustering algorithm where we apply spectral clustering to an estimated distance matrix. The distance matrix $\widehat{D} \in \mathbb{R}^{m \times m}$ is computed on all pairs of graphs using the defined estimator function (2), that is $\widehat{D}_{ij} = d(G_i, G_j)$. Unlike the standard Laplacian based spectral clustering, which is applicable for adjacency or similarity matrices, we use the method suggested by Mukherjee et al. (2017) that

computes the $K$ leading eigenvectors of $\widehat{D}$ (corresponding to the $K$ smallest eigenvalues in magnitude) and applies k-means clustering to the rows of the eigenvector matrix resulting in $K$ number of clusters. We refer to this distance based clustering algorithm as DSC, described in Algorithm 1 of Appendix. To derive the statistical consistency of DSC, we consider the problem of clustering $m$ random graphs of potentially different sizes, each sampled from one of $K$ graphons. We establish the consistency in Theorem 1 by proving that the number of misclustered graphs $\rightarrow 0$ asymptotically.

**Theorem 1 (Consistency of DSC)** *Consider $K$ graphons satisfying Assumptions 1–3, and $m$ random graphs $G_1, \ldots, G_m$, each sampled from one of the $K$ graphons (assume there is at least one graph from each graphon). Define the distance matrix $D \in \mathbb{R}^{m \times m}$ such that $D_{ij} = \|w_i - w_j\|_{L_2}$ where $w_i$ and $w_j$ are the graphons from which $G_i$ and $G_j$ are generated. Let $n$ be the size of the smallest graph, and $\gamma$ be the $K$-th smallest eigenvalue value of $D$ in magnitude. As $n \rightarrow \infty$, if $n_0$ is chosen such that $\frac{m^2 n_0^2 \log n}{n} \rightarrow 0$, then DSC misclusters at most $\mathcal{O}\left(\frac{m^3}{\gamma^2 n_0^2}\right)$ graphs w.h.p.*

*Proof sketch.* The proof, given in Appendix A.3, uses the Davis-Kahan spectral perturbation theorem to bound the error in terms of $\|\widehat{D} - D\|_F$, which is further bounded using Proposition 1. □

While the number of misclustered graphs seem to depend on $m^3$, we note that there is an inverse dependence on $\gamma^2$ which has dependence on $m$ (see Corollary 1 that illustrates it for a specific case). Moreover, our focus is on the setting where $m = \mathcal{O}(1)$ and $n, n_0 \rightarrow \infty$, in which case, the error asymptotically vanishes. It is natural to wonder whether the dependence on $m$ and $n_0$ is tight in the above bounds. Currently, we do not know the optimal rates, but deriving this would be difficult due to the strong dependency of entries in $\widehat{D}$ and slow rate of convergence of the graph distance in Proposition 1. The presence of $\gamma$ in the above clustering error bound makes Theorem 1 less interpretable. Hence, we also consider the specific case of $K = 2$ (two graphons) in the following result, along with the assumption that equal number of graphs are generated from both graphons.

**Corollary 1** *Let $w \neq w'$ be two graphons satisfying Assumptions 1–3, and $m$ is a bounded even number. Assume that equal number of graphs are generated from $w$ and $w'$. For any $n_0$ and large enough constant $C$ such that $\|w - w'\|_{L_2} \geq C \frac{m}{n_0}$ and $\frac{m^2 n_0^2 \log n}{n} \rightarrow 0$ as $n \rightarrow \infty$, the number of graphs misclustered by Algorithm 1 goes to zero w.h.p.*

The corollary implies that given the observed graphs are large enough, and if the choice of $n_0$ is relatively small, $n_0 \ll \sqrt{n/\log n}$, and the graphons are $\Omega(\frac{1}{n_0})$ apart in $L_2$-distance, then the clustering is consistent. Intuitively, it can be understood that if we condense large graphs to a small representation (small $n_0$), then the clusters can be identified only if the models are quite dissimilar.

## 3.2 SIMILARITY BASED SEMI-DEFINITE PROGRAMMING (SSDP)

We propose another algorithm for clustering $m$ graphs based on similarity between pairs of graphs. The pairwise similarity matrix $\widehat{S} \in \mathbb{R}^{m \times m}$ is computed by applying Gaussian kernel on the distance between the graphs, that is $\widehat{S}_{ij} = \exp\left(-\frac{d(G_i, G_j)}{\sigma_i \sigma_j}\right)$, where $\sigma_1, \ldots, \sigma_n$ are parameters. For theoretical analysis, we assume $\sigma_1 = \ldots = \sigma_n$ is fixed, but in experiments, the parameters are chosen adaptively. We use the following semi-definite program (SDP) (Yan et al., 2018; Perrot et al., 2020) to find membership of the observed graphs. Let $X \in \mathbb{R}^{m \times m}$ be the normalised clustering matrix, that is $X_{ij} = 1/|\mathcal{C}|$ if $i$ and $j$ belong to cluster $\mathcal{C}$, and 0 otherwise. Then, SDP for estimating $X$ is:

$$\max_X \operatorname{trace}(\widehat{S}X) \qquad \text{s.t. } X \geq 0, \ X \succeq 0, \ X\mathbf{1} = \mathbf{1}, \ \operatorname{trace}(X) = K, \qquad (4)$$

where $X \geq 0$, $X \succeq 0$ ensure that $X$ is a non-negative, positive semi-definite matrix, and $\mathbf{1}$ denotes the vector of all ones. We denote the optimal $X$ from the SDP as $\widehat{X}$. Once we have $\widehat{X}$, we apply standard spectral clustering on $\widehat{X}$ to obtain a clustering of the graphs. We refer to this algorithm as SSDP, described in Algorithm 2 of Appendix. We present strong consistency result for SSDP below.

**Theorem 2 (Consistency of SSDP)** *Consider $K$ graphons, $w_1, \ldots, w_K$, satisfying Assumptions 1–3, and $m$ random graphs, each sampled from one of the $K$ graphons. Let $n$ be the size of the smallest graph. As $n \rightarrow \infty$, if $n_0$ is chosen such that $\frac{m^2 n_0^2 \log n}{n} \rightarrow 0$ and $\min_{l \neq l'} \|w_l - w_{l'}\|_{L_2} = \Omega\left(\frac{m}{n_0}\right)$, then the number of graphs misclustered by SSDP is zero w.h.p.*

*Proof sketch.* The proof in Appendix A.4 adapts Perrot et al. (2020, Proposition 1) to the present setting and combines it with Proposition 1 to derive the stated condition for zero error. □

Theorem 2 is slightly stronger than Theorem 1, or Corollary 1, since SSDP achieve a zero clustering error for large enough graphs. This theoretical merit of SDP over spectral clustering is known in the statistics literature. Similar to Corollary 1, the choice of $n_0$ is important such that it does not violate the minimum $L_2$-distance condition in the theorem to ensure consistency.

**Remark on the knowledge of $K$.** Above discussions assume that the number of clusters $K$ is known, which is not necessarily the case in practice. To tackle this issue, one can estimate $K$ using Elbow method (Thorndike, 1953) or approach from Perrot et al. (2020),and then use it as input in our algorithms, DSC and SSDP. One can modify the SDP (4) and Theorem 2 to the case where $K$ is adaptively estimated. However, we found the corresponding algorithm, adapted from Perrot et al. (2020), to be empirically unstable in the present context. Hence, the knowledge of $K$ is assumed in the following experiments, which also allows the efficacy of the proposed algorithms and graph distance to be evaluated without the error induced by incorrect estimation of $K$.

## 3.3 EXPERIMENTAL ANALYSIS

In this section, we evaluate the performance of our algorithms DSC and SSDP, both in terms of accuracy and computational efficacy. We measure the performance of the algorithms in terms of error rate, that is, the fraction of misclustered graphs by using the source of the graphs as labels. Since clustering provides labels up to permutation, we use the Hungarian method (Kuhn, 1955) to match the labels. The performance can also be measured in terms of Adjusted Rand Index (results in Appendix C.3). We use simulated and real datasets for evaluation and obtain all our experimental results using Tesla K80 GPU instance with 12GB memory from Google. Code available in Github.

**Simulated data.** We generate graphs of varied sizes from four graphons, $W_1(u, v) = uv$, $W_2(u, v) = \exp\{-\max(u, v)^{0.75}\}$, $W_3(u, v) = \exp\{-0.5 * (\min(u, v) + u^{0.5} + v^{0.5})\}$ and $W_4(u, v) = |u - v|$. The simulated graphs are dense and the graph sizes are controlled to study how algorithms scale. Their corresponding $L_2$ distances between pairs of graphons is shown later in Figure 3 and the heatmap of the graphons are visualised in Figure 4 in Appendix.

**Real data.** We use datasets from two contrasting domains: small molecule datasets from Bioinformatics and large network datasets from Social Networks. We use *Proteins* (Borgwardt et al., 2005), *KKI* (Pan et al., 2016), *OHSU* (Pan et al., 2016) and *Peking_1* (Pan et al., 2016) datasets from Bioinformatics, and *Facebook_Ct1* (Oettershagen et al., 2020), *Github_Stargazers* (Rozemberczki et al., 2020), *Deezer_Ego_Nets* (Rozemberczki et al., 2020) and *Reddit_Binary* (Yanardag & Vishwanathan, 2015) datasets from Social Networks. We sub-sample a few graphs from each dataset by setting a minimum number of nodes to validate clustering small number of large graphs (small $m$, large $n$). The number (#graphs) and size (#nodes) of the graphs are listed in Figure 1 tables. We consider all combinations of the datasets for three and four clusters in both the domains separately.

**Choice of $n_0$ and $\sigma_i$.** As noted in our algorithms DSC and SSDP, $n_0$ is an input parameter. Theorems 1 and 2 show the choice of $n_0 = \mathcal{O}(\sqrt{n/\log n})$. Hence, we set $n_0 = \sqrt{n/\log n}$ where $n$ is the minimum number of nodes. In Appendix C.2, we use simulated data to show that the above choice of $n_0$ is reasonable (if not the best) for both DSC and SSDP. Furthermore, the similarity matrix $\widehat{S}$ in SSDP is computed using parameters $\sigma_i$ and we set $\sigma_i = d(G_i, G_{5nn})$ where $G_{5nn}$ is the fifth nearest neighbour of $G_i$. Hence, apart from knowledge of $K$, our algorithms are parameter-free.

**Performance comparison with existing methods.** We compare our algorithms with a range of approaches for measuring similarity or distance among multiple networks. Most methods discussed below provide a kernel or distance matrix to which we apply spectral clustering to obtain the clusters:

1) Network Clustering based on Log-Moments (NCLM) is an embedding based clustering strategy for different sized graphs (Mukherjee et al., 2017) that is based on network statistics called log moments. Log moments for a graph with adjacency matrix $A$ and number of nodes $n$ is obtained by $(\log(m_1(A)), \log(m_2(A)), \ldots, \log(m_J(A)))$ where $m_i(A) = \text{trace}(A/n)^i$ and $J$ is a parameter.

2) Wasserstein Weisfeiler-Lehman Graph Kernels (WWLGK) is a recent graph kernel that is based on the Wasserstein distance between the node feature vector distributions of two graphs proposed by Togninalli et al. (2019).

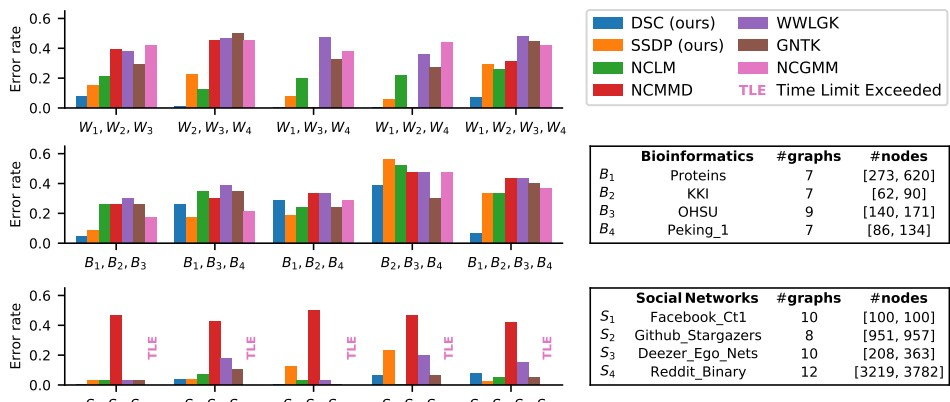

Figure 1: Evaluation of DSC and SSDP with other methods. **(row 1)** Results on simulated data. **(rows 2 and 3)** Results on real data from Bioinformatics and Social Networks, respectively. DSC outperforms in majority of the cases. Tables in rows 2 and 3 show details of the considered datasets.

3) Graph Neural Tangent Kernel (GNTK) is another graph kernel that describes infinitely wide graph neural networks derived by Du et al. (2019). Both WWLGK and GNTK provide state-of-the-art performance in graph classification with GNTK outperforming most graph neural networks.

4) Network Clustering algorithm based on Maximum Mean Discrepancy (NCMMD) considers a graph metric (MMD) to cluster the graphs. MMD distance between random graphs is proposed as an efficient test statistic for random dot product graphs (Agterberg et al., 2020). We compute MMD between the graphs that are represented by latent finite dimensional embedding called spectral adjacency embedding with the dimension $r$ as a parameter.

5) In Network Clustering algorithm based on Graph Matching Metric (NCGMM), we match two graphs of different sizes by appending null nodes to the small graph (Guo et al., 2019) and compute Frobenius norm between the matched graphs as their distance. Although the graph metrics (MMD and graph matching) are for different purposes, we evaluate their efficacy in the context of clustering.

The parameters in the algorithms are $n_0$ and $\sigma_i$ in DSC and SSDP, $J$ in NCLM, number of iterations ($\#itr$) to perform in WWLGK, number of layers ($\#layer$) in graph neural networks for GNTK, $r$ in NCMMD and none in NCGMM. We fix $n_0$ in our algorithms using the theoretical bound and $\sigma_i$ is set adaptively as discussed, whereas use grid search to tune the parameters for other algorithms.

**Evaluation on simulated data.** We sample 10 graphs of varied sizes between 50 and 100 nodes from each of the four graphons in Figure 4, and perform the experiments by considering all combinations of three and four clusters of the graphons. Based on the theoretical bound, $n_0$ is fixed to 5 as $n = 50$. We report the performance for $J = 8$, $r = 3$, $\#itr = 1$ and $\#layer = 2$ as these produce the best results. The first row of Figure 1 shows the average performance of the algorithms computed over 5 independent runs. We observe that our algorithm DSC outperforms all the other algorithms, achieving nearly zero error in all cases, and SSDP also performs competitively by standing second or third best. The graph kernels, WWLGK and GNTK, and the graph metric based method NCGMM typically do not perform well. NCMMD either performs very well or quite poorly. We sample small graphs since otherwise GNTK cannot run due to memory requirement for dense large graphs and NCGMM has high computation time. Appendix C.4 includes evaluation of the algorithms except GNTK and NCGMM on larger graphs, where we observe similar behaviour.

**Evaluation on real data.** We consider all combinations of three and four clusters of both Bioinformatics and Social Networks separately. The second and third rows of Figure 1 show the performance with $n_0 = 30$, $J = 8$, $r = 3$, $\#itr = 1$ and $\#layer = 2$, and the upper limit of 7200 seconds (2 hours) as running time of algorithms. We observe DSC outperforms other algorithms by a large margin in majority of the combinations, while in the other combinations like {Proteins,KKI,Peking_1}, DSC performs well with a very small margin to the best performing one. Although NCLM and GNTK compare favorably in Social Networks datasets, they typically have high error rate in Bioinformatics or simulated datasets, suggesting that they could be well suited for large networks, whereas

DSC is more versatile and suitable for all networks. SSDP performs moderately on real data, but it achieves the smallest error in some cases, implying that it is suited for certain types of networks.

**Computation time comparison.** Figure 2 shows the time (measured in seconds) taken by each algorithm for four clusters case, plotted in log scale. Appendix C.5 illustrates similar behavior in three clusters case as well. Our algorithms, DSC and SSDP, perform competitively with respect to time as well. In addition, it scales effectively for large graphs unlike other algorithms. It is worth noting that although NCLM takes lesser time than DSC and SSDP for small graphs, it takes longer for large social networks datasets, thus favoring our methods in terms of both accuracy and scalability. Graph matching based algorithm, NCGMM, has severe scalability issue showing the inapplicability of such methods to learning problems. We also evaluate the scalability of all algorithms by measuring the time for clustering different sets of varied sized graphs from graphons $W_1, W_2, W_3$ and $W_4$. Detailed discussion on high scalability of DSC and SSDP is given in Appendix C.6.

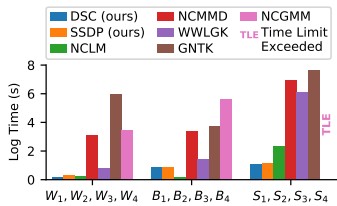

Figure 2: Computation time

## 4 GRAPH TWO-SAMPLE TESTING

Inspired by the remarkable performance of our graph distance (2) in clustering, we analyse its applicability in graph two-sample testing. Two-sample testing is usually studied in the large sample case $m \to \infty$, and several nonparametric tests are known that could also be applied to graphs. However, it is also relevant to study the small sample setting in graphs, particularly $m = 2$, that is, whether two large graphs are statistically identical or not (Ghoshdastidar et al., 2020; Agterberg et al., 2020).

We consider the following formulation of the graph two-sample problem, stated under the assumption that the graphs are sampled from graphons. Given two random graphs, $G_1$ sampled from some model (here, graphon $w_1$), and $G_2$ from another model $w_2$, the goal is to determine which of the following hypothesis is true: $H_0 : \{w_1 = w_2\}$ or $H_a : \{w_1 \neq w_2 : \|w_1 - w_2\|_{L_2} \geq \phi\}$ for some $\phi > 0$. Existing works consider alternative random graph model, such as inhomogeneous Erdős-Rényi models or random dot product graph models, which are more restrictive. The condition $\phi > 0$ is necessary if one only has access to finitely many independent samples (Ghoshdastidar et al., 2020). A two-sample test $T$ is a binary function of the given samples such that $T = 1$ denotes that the test rejects the null hypothesis $H_0$ and $T = 0$ implies that the test rejects the alternate hypothesis $H_a$. The goodness of a two-sample test is measured in terms of the Type-I and Type-II errors, which denote the probabilities of incorrectly rejecting the null and alternate hypotheses, respectively. The goal of this section is to show that one can construct a test $T$ that has arbitrarily small Type-I and Type-II errors. For this purpose, we consider the test

$$T : \mathbb{I}\{d(G_1, G_2) \geq \xi\} \tag{5}$$

for some $\xi > 0$, where $\mathbb{I}\{\cdot\}$ is the indicator function and $d(G_1, G_2)$ is the proposed graph distance for some choice of integer $n_0$. We state the following theoretical guarantee for the test $T$.

**Theorem 3** *Assume that the graphons $w_1, w_2$ satisfy Assumptions 1–3, and let the graphs $G_1 \sim w_1$ and $G_2 \sim w_2$ have at least $n$ nodes. As $n \to \infty$, there is a choice of $\xi$ such that the Type-I and Type-II errors of the test $T$ in (5) go to 0 if $\frac{n_0^2 \log n}{n} \to 0$ and $\phi \geq \frac{C}{n_0}$, where the constant $C$ depends only on the Lipschitz constants.*

Theorem 3 shows that the test $T$ in (5) can distinguish between any pair of graphons that have separation $\|w_1 - w_2\|_{L_2} = \Omega(1/n_0)$ with arbitrarily small error, if the graphs are large enough.

**Empirical analysis.** We validate the consistency result in Theorem 3 by computing power of the proposed test $T$, which measures the probability of rejecting the null hypothesis $H_0$. Intuitively, power of the test for graphs sampled from same graphons should be small (close to a pre-specified significance level) since $H_0$ must not be rejected, whereas, it should be close to 1 for graphs sampled from different graphons. As known in the testing literature, theoretical threshold, $\xi$ in (5), is typically conservative in practice and the rejection/acceptance is decided based on $p$-values, computed using bootstrap samples. To this end, we follow the bootstrapping strategy in Ghoshdastidar & von Luxburg (2018, Boot-ASE algorithm). We also compare the test $T$ by replacing $d(G_1, G_2)$ in 5

Figure 3: **(left)** $L_2$ distance between the graphons $W_1$, $W_2$, $W_3$ and $W_4$. **(other plots)** Average power of the test (5) for a graph pair of sizes 100 and 200, sampled from every pair of graphons.

with two other statistics, log moments (Mukherjee et al., 2017) and MMD, an efficient test statistics for random dot product graphs (Agterberg et al., 2020). We perform the experiment by sampling graphs $G_1 \sim w_1$ and $G_2 \sim w_2$ of size $n$ and $2n$, respectively, where $w_1$ and $w_2$ are chosen from graphons $W_1, W_2, W_3, W_4$. The power of $T$ is computed for $n = 100$ (thus $n_0 = 10$ from the theoretical bound) and the significance level 0.05, averaged over 500 trials of bootstrapping 100 samples generated from all pairs of graphons. The plots in Figure 3 show the average power of $T$ with our proposed distance, log moments and MMD as $d(G_1, G_2)$, respectively. From the results, it is clear that $T$ using our distance can distinguish between graphons that are quite close too (smallest $L_2$ distance for $W_2$ and $W_3$), whereas, other test statistics are weak as log moments statistic accepts $H_0$ even when it is wrong (see $W_1$ and $W_4$) and MMD based test rejects it strongly almost always (see diagonal). Appendix C shows similar result for small and large $n$, and evaluation on real datasets.

## 5   CONCLUSION

There has been significant progress in learning on complex data, including network-valued data. However, much of the theoretical and algorithmic development have been in large sample problems, where one has access to $m \to \infty$ independent samples. Practical applications of network-valued data analysis often leads to small sample, large graph problems—a setting where the machine learning literature is quite limited. Inspired by graph limits and high-dimensional statistics, this paper proposes a simple graph distance (2) based on non-parametric graph models (graphons).

Sections 3–4 demonstrate that the proposed graph distance leads to provable and practically effective algorithms for clustering (DSC and SSDP) as well as two-sample testing (5). Extensive empirical studies on simulated and real data show that the clustering based on the graph distance (2) outperforms methods based on more complex graph similarities or metrics, both in terms of accuracy and scalability. Figures 1–2 show that DSC achieves best performance for both small dense graphs (simulated graphons) as well as large sparse graphs (social networks). On the other hand, popular machine learning approaches—graph kernels or graph matching—can be computationally expensive in large graphs and their performance may not improve as $n \to \infty$, see WWLGK in Figure 7.

Statistical approaches, such as the proposed clustering algorithms and two-sample test, show better performance on large graphs (Figures 1, 3 and Appendix C.4). Theorems 1–3 theoretically support this observation by showing consistency of the clustering and testing methods in the limit of $n \to \infty$. The theoretical results, however, hinge on Assumptions 1–3. We remark that such smoothness and equivalence assumptions could be necessary for meaningful non-parametric approaches, which is also supported by the graph testing and graphon estimation literature. Further insights about the necessity of smoothness assumptions would aid in theoretical and algorithmic development.

The poor performance of graph kernels and graph matching in clustering and small sample problems calls for further studies on these methods, which have shown success in network classification. Fundamental research, combining graphon based approaches and kernels, could lead to improved techniques. Instead of sorting-and-smoothing graphon estimator, other histogram based methods that reduce graphons to a fixed size by identifying block structures and averaging (Olhede & Wolfe, 2014) can be explored. However, the distance defined on such reduced graphons require alignment of the blocks between the graphons. While such methods do not require Assumption 2 and allow sparse graphs, the computationally inefficient alignment step pose a practical challenge. Algorithmic modifications, such as estimation of $K$, would be also useful in practice.

## 6 ACKNOWLEDGEMENTS

This work has been supported by the German Research Foundation (Research Training Group GRK 2428) and the Baden-Württemberg Stiftung (Eliteprogram for Postdocs project "Clustering large evolving networks"). The authors thank the International Max Planck Research School for Intelligent Systems (IMPRS-IS) for supporting Leena Chennuru Vankadara.

## 7 ETHICS STATEMENT

The proposed clustering algorithms for network-valued data can naturally be evaluated on real-world data, but no social interpretation about the results will be drawn. Moreover, development of fair clustering algorithms is not the focus of this project.

## 8 REPRODUCIBILITY STATEMENT

The assumptions for the theory are stated clearly in Assumptions 1–3 and all the theoretical results, Proposition 1, Theorems 1-3, Corollary 1, are proved in detail in Appendix A. The implementation of the considered algorithms with steps to reproduce the results is available in Github - https://tinyurl.com/3ycmmj4u. Datasets used in the experiments are public and the links are provided in `download_datasets` function of `utils.py`. The experimental results can be reproduced by following `graph_clustering.ipynb`.

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

# A  PROOFS OF THEORETICAL RESULTS

We discuss the proofs of Proposition 1 and Theorems 1–3 with supporting lemmas in this section.

## A.1  PROPOSITION 1

The distance function defined in (2) estimates the $L_2$-distance between graphons that are continuous. To prove this, we introduce a method to discretize the continuous graphons in the following so that it is comparable with the transformed graphs described in the graph distance estimator (2).

**Graphon discretization.** We discretize the graphon by applying piece-wise constant function approximation that is inspired from Chan & Airoldi (2014), similar to the graph transformation. More precisely, any continuous graphon $w$ is discretized to a matrix $W$ of size $n_0 \times n_0$ with

$$W_{ij} = \frac{1}{1/n_0^2} \int_0^{\frac{1}{n_0}} \int_0^{\frac{1}{n_0}} w(x + \frac{i}{n_0}, y + \frac{j}{n_0}) \, \mathrm{d}x \, \mathrm{d}y \qquad (6)$$

We make Assumptions 1–3 to derive the concentration bound stated in Proposition 1. The proof structure is as follows:

1. We bound the point-wise deviation of graphon with its discretized correspondence using Lipschitzness assumption (Lemma 1).

2. We derive the error bound between the $L_2$-distance and Frobenius norm of the discretized graphons using Lemma 1 (Lemma 2).

3. We establish a relation between Frobenius norm of the histograms of graphons and graphs (Lemma 3).

4. Finally, we prove Proposition 1 by combining Lemmas 2 and 3.

**Lemma 1 (Lipschitz condition)** *For any graphon $w$ and corresponding discretization $W$, define a piecewise constant function $\overline{w(x,y)} = W_{ij}$ where $x \in [\frac{i}{n_0}, \frac{i}{n_0} + \frac{1}{n_0}]$ and $y \in [\frac{j}{n_0}, \frac{j}{n_0} + \frac{1}{n_0}]$. Using the Lipschitz continuous assumption, we have*

$$\left| \overline{w(x,y)} - w(x,y) \right| \leq \frac{2\sqrt{2}L}{n_0}, \qquad (7)$$

*where $L$ is the Lipschitz constant in Assumption 1.*

**Proof of Lemma 1 (Lipschitz condition).** We use Assumption 1 on Lipschitzness to prove this lemma. The following holds for a graphon $w$ with Lipschitz constant $L$,

$$\left| w(x + \frac{i}{n_0}, y + \frac{j}{n_0}) - w(x,y) \right| \leq L \sqrt{\frac{i^2}{n_0^2} + \frac{j^2}{n_0^2}} \leq \frac{L\sqrt{2}}{n_0} \qquad (8)$$

We prove Lemma 1 using (8) and the definition of $\overline{w(x,y)} = W_{ij}$ where $x \in [\frac{i}{n_0}, \frac{i}{n_0} + \frac{1}{n_0}]$ and $y \in [\frac{j}{n_0}, \frac{j}{n_0} + \frac{1}{n_0}]$,

$$\left| \overline{w(x,y)} - w(x,y) \right| = \left| \overline{w(x,y)} - w(x,y) \pm w(x + \frac{i}{n_0}, y + \frac{j}{n_0}) \right|$$

$$\leq \left| \overline{w(x,y)} - w(x + \frac{i}{n_0}, y + \frac{j}{n_0}) \right| + \left| w(x + \frac{i}{n_0}, y + \frac{j}{n_0}) - w(x,y) \right| = \frac{2\sqrt{2}L}{n_0} \square$$

**Lemma 2 (Error bound of discretization)** *For two graphons $w_1$ and $w_2$, the error bound between the $L_2$-distance and the Frobenius norm of the corresponding discretized graphons $W_1$ and $W_2$ satisfies*

$$\left| \| w_1 - w_2 \|_{L_2} - \frac{1}{n_0} \| W_1 - W_2 \|_F \right| \leq \frac{4\sqrt{2}L}{n_0}. \qquad (9)$$

**Proof of Lemma 2 (Error bound of the approximation).** Lemma 1 is used to prove this lemma. Let $L_1$ and $L_2$ be the Lipschitz constants of $w_1$ and $w_2$.

$$\| w_1 - w_2 \|_{L_2}^2 = \int_0^1 \int_0^1 (w_1(x,y) - w_2(x,y))^2 \, \mathrm{d}x \, \mathrm{d}y$$

$\pm \overline{w_1(x,y)}$ and $\pm \overline{w_2(x,y)}$ within the square, expand and apply Lipschitzness condition from Lemma 1

$$\leq \frac{8L_1^2}{n_0^2} + \frac{8L_2^2}{n_0^2} + \frac{16L_1L_2}{n_0^2} + \sum_{k=0}^{n_0}\sum_{l=0}^{n_0} \frac{1}{n_0^2}\Big((W_1)_{kl} - (W_2)_{kl}\Big)^2$$

$$+ \frac{4\sqrt{2}(L_1+L_2)}{n_0}\sum_{k=0}^{n_0}\sum_{l=0}^{n_0}\frac{1}{n_0^2}\Big|(W_1)_{kl} - (W_2)_{kl}\Big|$$

$$\leq \Big(\frac{2\sqrt{2}(L_1+L_2)}{n_0}\Big)^2 + \frac{1}{n_0^2}\|W_1 - W_2\|_F^2 + \frac{4\sqrt{2}(L_1+L_2)}{n_0^3}\|W_1 - W_2\|_1$$

$$\overset{(a)}{\leq} \Big(\frac{2\sqrt{2}(L_1+L_2)}{n_0} + \frac{1}{n_0}\|W_1 - W_2\|_F\Big)^2 \tag{10}$$

$(a): \|x\|_1 \leq \sqrt{n}\,\|x\|_F$

Similarly, by $\pm w_1(x,y)$ and $\pm w_2(x,y)$ to $\frac{1}{n_0^2}\|W_1 - W_2\|_F^2$ and applying Lipschitzness condition from Lemma 1 we get,

$$\frac{1}{n_0^2}\|W_1 - W_2\|_F^2 \leq \Big(\frac{2\sqrt{2}(L_1+L_2)}{n_0} + \|w_1 - w_2\|_{L_2}\Big)^2 \tag{11}$$

Combining (10) and (11), we prove

$$\Big|\| w_1 - w_2 \|_{L_2} - \frac{1}{n_0}\|W_1 - W_2\|_F\Big| \leq \frac{2\sqrt{2}(L_1+L_2)}{n_0} \overset{(b)}{\leq} \frac{4\sqrt{2}L}{n_0}, \quad \text{(b)}: L = \max\{L_1, L_2\}\,\square$$

We derive the following relation between histogram of graphs and graphons by adapting lemmas from Chan & Airoldi (2014) for our problem and the error bound of discretization (9).

**Lemma 3** *Let $G_1 \sim w_1$ and $G_2 \sim w_2$ have respective graph transformations $A_1$ and $A_2$. Let $W_1$ and $W_2$ be the corresponding discretized graphons of $w_1$ and $w_2$, respectively. As $n \to \infty$ and $\frac{n_0^2 \log n}{n} \to 0$, then for any $\epsilon > 0$,*

$$\big|\| A_1 - A_2 \|_F - \| W_1 - W_2 \|_F\big| \leq 4\,\epsilon \tag{12}$$

*with probability converging to* 1.

**Proof of Lemma 3.** The proof of this lemma is inspired from Chan & Airoldi (2014). For $i = \{1,2\}$, let matrix $W_i$ of size $n_0 \times n_0$ be the discretized graphon $w_i$ and let matrix $\widehat{A}_i$ of size $n_0 \times n_0$ be another transformation of graph $G_i$ based on the true permutation $\widehat{\sigma}_i$, that is, $\widehat{\sigma}_i$ denotes the ordering of the graph $G_i$ based on the corresponding graphon $w_i$. In other words, the discretized graphon $W_i$ is the expectation of $\widehat{A}_i$. The reordered graph is denoted by $G_i^{\widehat{\sigma}}$ and then $\widehat{A}_i$ is obtained by,

$$(\widehat{A}_i)_{kl} = \frac{1}{h^2}\sum_{k_1=0}^{h}\sum_{l_1=0}^{h}(G^{\widehat{\sigma}_i})_{kh+k_1, lh+l_1} \quad \text{where } h = \lfloor\frac{n}{n_0}\rfloor \text{ and } \lfloor\cdot\rfloor \text{ is the floor function.}$$

We bound $\|A_1 - A_2\|_F$ using $\widehat{A}_1, \widehat{A}_2, W_1$ and $W_2$ as,

$$\|A_1 - A_2\|_F \leq \|A_1 - \widehat{A}_1\|_F + \|A_2 - \widehat{A}_2\|_F + \|\widehat{A}_1 - \widehat{A}_2\|_F$$

$$\leq 2\max_{i=\{1,2\}}\|A_i - \widehat{A}_i\|_F + \|\widehat{A}_1 - \widehat{A}_2\|_F$$

$$\leq 2\max_{i=\{1,2\}}\|A_i - \widehat{A}_i\|_F + 2\max_{i=\{1,2\}}\|\widehat{A}_i - W_i\|_F + \|W_1 - W_2\|_F \tag{13}$$

We have the following for all $i$ using Assumption 2 on Lipschitzness of the expected degree function $g(u)_i$ of graphon $w_i$ and Lemma 3 of Chan & Airoldi (2014),

$$\mathbb{E}[\|A_i - \widehat{A}_i\|_F^2] \leq \frac{n_0^4}{n_i^2} \left(2 + 4C_i^2 L_i^2 \frac{\log n_i}{n_i}\right) + n_0^2 \left(4C_i^2 L_i^2 \frac{\log n_i}{n_i}\right) \; ; C_i \text{ depends on Lipschitz constants of } g(u)_i$$

$$\leq 2\frac{n_0^4}{n^2} + \frac{n_0^4}{n^2} 4C_i^2 L_i^2 \frac{\log n}{n} + n_0^2 4C_i^2 L_i^2 \frac{\log n}{n} = \mathcal{O}\left(n_0^2 \frac{\log n}{n}\right) \tag{14}$$

Applying Markov's inequality to bound the probability $\mathbb{P}\left(\max_{i=\{1,2\}} \|A_i - \widehat{A}_i\|_F^2 \geq \epsilon_1^2\right)$ using (14),

$$\mathbb{P}\left(\max_{i=\{1,2\}} \|A_i - \widehat{A}_i\|_F^2 \geq \epsilon_1^2\right) \overset{(a)}{\leq} \sum_{i=\{1,2\}} \frac{\mathbb{E}\left[\|A_i - \widehat{A}_i\|_F^2\right]}{\epsilon_1^2} \overset{(14)}{=} \mathcal{O}\left(\frac{n_0^2 \log n}{\epsilon_1^2 n}\right) \; ; (a) : \text{Union bound}$$

$$\tag{15}$$

Thus asymptotically, as $n \to \infty$ and $\frac{n_0^2 \log n}{n} \to 0$, for all $i$ and any $\epsilon_1 > 0$, $\|A_i - \widehat{A}_i\|_F < \epsilon_1$ with probability converging to 1.

From Lemma 4 of Chan & Airoldi (2014), we have the following for all $i$,

$$\mathbb{E}[\|\widehat{A}_i - W_i\|_F^2] \leq \frac{n_0^4}{n_i^2} \leq \frac{n_0^4}{n^2} \tag{16}$$

Applying Markov's inequality to bound the probability $\mathbb{P}\left(\max_{i=\{1,2\}} \|\widehat{A}_i - W_i\|_F^2 \geq \epsilon_2^2\right)$ using (16),

$$\mathbb{P}\left(\max_{i=\{1,2\}} \|\widehat{A}_i - W_i\|_F^2 \geq \epsilon_2^2\right) \overset{(a)}{\leq} \sum_{i=\{1,2\}} \frac{\mathbb{E}\left[\|\widehat{A}_i - W_i\|_F^2\right]}{\epsilon_2^2} \overset{(16)}{\leq} \frac{2n_0^4}{\epsilon_2^2 n^2} \; ; (a) : \text{Union bound}$$

$$\tag{17}$$

Again asymptotically as $n \to \infty$, for all $i$ and any $\epsilon_2 > 0$, $\|\widehat{A}_i - W_i\|_F < \epsilon_2$ with probability converging to 1. Lets assume $\epsilon_2 = \epsilon_1 = \epsilon$. Substituting (15) and (17) in (13),

$$\|A_1 - A_2\|_F \leq 4\epsilon + \|W_1 - W_2\|_F \tag{18}$$

with probability converging to 1 as $n \to \infty$ and $\frac{n_0^2 \log n}{n} \to 0$.

The lower bound can similarly be obtained,

$$\|A_1 - A_2\|_F \geq \|W_1 - W_2\|_F - 2 \max_{i=\{1,2\}} \|\widehat{A}_i - W_i\|_F - 2 \max_{i=\{1,2\}} \|\widehat{A}_i - A_i\|_F$$

$$\geq \|W_1 - W_2\|_F - 4\epsilon \tag{19}$$

with probability converging to 1 as $n \to \infty$ and $\frac{n_0^2 \log n}{n} \to 0$.

Thus, $|\|A_1 - A_2\|_F - \|W_1 - W_2\|_F| \leq 4\epsilon$ satisfy for any $\epsilon > 0$ with probability converging to 1 as $n \to \infty$ and $\frac{n_0^2 \log n}{n} \to 0$ from equations (18) and (19). $\qquad\square$

**Proof of Proposition 1 (Graph distance is consistent).** Proposition 1 immediately follows from Lemmas 2 and 3 after a simple decomposition step as shown below.

$$|\|w_1 - w_2\|_{L_2} - d(G_1, G_2)| = \left|\|w_1 - w_2\|_{L_2} - d(G_1, G_2) \pm \frac{1}{n_0}\|W_1 - W_2\|_F\right|$$

$$\leq \left|\|w_1 - w_2\|_{L_2} - \frac{1}{n_0}\|W_1 - W_2\|_F\right| + \left|\frac{1}{n_0}\|W_1 - W_2\|_F - d(G_1, G_2)\right|$$

$$\overset{2,3}{\leq} \frac{4\sqrt{2}L}{n_0} + \frac{4\epsilon}{n_0} = \mathcal{O}\left(\frac{1}{n_0}\right) \text{ holds for any } \epsilon > 0 \text{ as } n \to \infty \text{ and } \frac{n_0^2 \log n}{n} \to 0 . \square$$

## A.2 PROPOSITION 1 FOR SPARSE GRAPHS

To adapt the dense graph model, graphon, to sparse graphs, the sampling procedure is modified to $G_{ij}|U_i, U_j \sim Bernoulli\left(\rho\, w(U_i, U_j)\right)$ where $\rho$ depends only on $n$. With this modification, equation 14 will be changed to

$$\mathbb{E}[\|A_i - \widehat{A}_i\|_F^2] \le 2\frac{n_0^4}{n^2} + \frac{n_0^4}{n^2} 4C_i^2 L_i^2 \rho^2 \frac{\log n}{n} + n_0^2 4C_i^2 L_i^2 \rho \frac{\log n}{n} = \mathcal{O}\left(n_0^2 \rho \frac{\log n}{n}\right) \qquad (20)$$

and the proof involves modification to Lemma 2 of Chan & Airoldi (2014). We state the modified lemma and briefly sketch the proof below.

**Modified Lemma 2 of Chan & Airoldi (2014).** Let $\sigma(i)$ be the oracle permutation such that $U_{\sigma(1)} < U_{\sigma(2)} < \ldots < U_{\sigma(n)}$. Then, if $\left|\frac{\sigma(i)}{n} - \frac{\sigma(j)}{n}\right| < \frac{1}{6L_1 \rho}\sqrt{\frac{\log n}{n}}$ then

$$\left|d_{\sigma(i)} - d_{\sigma(j)}\right| < \sqrt{\frac{\log n}{n}} \qquad (21)$$

with probability at least $1 - 8\exp\left(-\frac{1}{18L_1^2}\frac{\log n}{\rho^2}\right)$. Conversely, if (21) holds with probability at least $1 - 8\exp\left(-\frac{1}{18L_1^2}\frac{\log n}{\rho^2}\right)$ then

$$\left|\frac{\sigma(i)}{n} - \frac{\sigma(j)}{n}\right| < \frac{1}{\rho}\sqrt{\frac{\log n}{n}}\left(\frac{1}{3L_1} + \frac{1}{3L_1 L_2} + \frac{1}{L_2}\right) \qquad (22)$$

with probability at least $1 - 40\exp\left(-\frac{1}{18L_1^2}\frac{\log n}{\rho^2}\right)$.

**Changes to Proof in Chan & Airoldi (2014).** Suppose $\left|\frac{\sigma(i)}{n} - \frac{\sigma(j)}{n}\right| < \frac{\delta}{\rho}$ for $\delta > 0$ and $\delta < \rho$. Then, $\mathbb{P}\left(\left|U_{\sigma(i)} - U_{\sigma(j)}\right| > 3\frac{\delta}{\rho}\right) \le 4\exp\left(-2n\frac{\delta^2}{\rho^2}\right)$. Consequently,

$$\mathbb{P}\left(\left|g(U_{\sigma(i)}) - g(U_{\sigma(j)})\right| > 3L_1\delta\right) \le \mathbb{P}\left(\left|U_{\sigma(i)} - U_{\sigma(j)}\right| > 3\frac{\delta}{\rho}\right) \le 4\exp\left(-2n\frac{\delta^2}{\rho^2}\right).$$

Following their proof, we have

$$\mathbb{P}\left(\left|d_{\sigma(i)} - d_{\sigma(j)}\right| > 6L_1\delta \mid U_{\sigma(i)}, U_{\sigma(j)}\right)$$
$$\le 4\exp\left(-\frac{9}{2}nL_1^2\delta^2\right) + 4\exp\left(-2n\frac{\delta^2}{\rho^2}\right) \le 8\exp\left(-2n\frac{\delta^2}{\rho^2}\right), \qquad (23)$$

when $\rho > \frac{2}{3L_1}$. Note that there is a small mistake in their proof when Hoeffding's inequality is applied, where a factor of $n^2$ is written in place of $n$. Putting $\delta = \frac{1}{6L_1}\sqrt{\frac{\log n}{n}}$ and considering $\rho = \Omega\left(\sqrt{\frac{\log n}{n}}\right)$ since $\rho$ depends on $n$ and $\delta < \rho$, we get

$$\mathbb{P}\left(\left|d_{\sigma(i)} - d_{\sigma(j)}\right| > \sqrt{\frac{\log n}{n}} \mid U_{\sigma(i)}, U_{\sigma(j)}\right) \le 8\exp\left(-\frac{1}{18L_1^2}\frac{\log n}{\rho^2}\right)$$

Converse can similarly be proved.

(20) can be derived by substituting the above changed lemma in their proof. Consequently, Proposition 1 still holds under this formulation, with slight change to the condition as $n \to \infty$ and $\frac{n_0^2 \rho \log n}{n} \to 0$ for $\rho = \Omega\left(\sqrt{\frac{\log n}{n}}\right)$. We will not consider sparsity $\rho$ for analysing consistency of our algorithms in the next sections.

A.3 DISTANCE BASED SPECTRAL CLUSTERING (DSC)

We make Assumptions 1–3 on the $K$ graphons to analyse the Algorithm 1. We establish the consistency of this algorithm by deriving the number of misclustered graphs $|\mathcal{M}|$ through the following steps.

1. We establish deviation bound between the estimated distance matrix $\widehat{D}$ and the ideal distance matrix $D$ (Lemma 4).
2. We formulate Davis-Kahan theorem in terms of the deviation bound using the result from Mukherjee et al. (2017) (Lemma 5).
3. We derive the number of misclustered graphs from Lemma 5.

As stated previously, $\widehat{D}$ in Algorithm 1 is an estimate of $D \in \mathbb{R}^{m \times m}$, where we define $D_{ij} = \|w_i - w_j\|_{L_2}$. Note that $D$ is a block matrix with rank $K$, since $D_{ij} = 0$ for all $G_i, G_j$ generated from same graphon, and equals the distance between the graphons $i$ and $j$ otherwise.

We derive the deviation bound for the distance matrix using Lemma 3 and the result is as follows.

**Lemma 4 (Distance deviation bound)** *As $n \to \infty$ and $\dfrac{n_0^2 \log n}{n} \to 0$, we establish*

$$\left\| \widehat{D} - D \right\|_F = \mathcal{O}\left( \frac{m}{n_0} \right) \tag{24}$$

*with probability converging to 1.*

**Proof of Lemma 4 (Distance deviation bound).** From Proposition 1 and the definitions of $\widehat{D}_{ij}$ and $D_{ij}$, it is easy to see that $\left| \widehat{D}_{ij} - D_{ij} \right| = \mathcal{O}\left( \dfrac{1}{n_0} \right)$ with probability converging to 1 as $n \to \infty$ and $\dfrac{n_0^2 \log n}{n} \to 0$.

Using Lemmas 2 and 3, and the definitions of $\widehat{D}_{ij}$ and $D_{ij}$, we have

$$
\begin{aligned}
\left| \widehat{D}_{ij} - D_{ij} \right| &= \left| \frac{1}{n_0} \|A_i - A_j\|_F - \|w_i - w_j\|_{L_2} \right| \\
&= \left| \frac{1}{n_0} \|A_i - A_j\|_F \pm \frac{1}{n_0} \|W_i - W_j\|_F - \|w_i - w_j\|_{L_2} \right| \\
&\leq \frac{1}{n_0} \left| \|A_i - A_j\|_F - \|W_i - W_j\|_F \right| + \left| \|w_i - w_j\|_{L_2} - \frac{1}{n_0} \|W_i - W_j\|_F \right| \\
&\overset{(12)}{\leq} \frac{4\epsilon}{n_0} + \frac{4\sqrt{2}L}{n_0} \qquad \text{with prob.} \to 1 \text{ asymptotically}
\end{aligned}
$$

Thus asymptotically, $\left| \widehat{D}_{ij} - D_{ij} \right| = \mathcal{O}\left( \dfrac{1}{n_0} \right)$ with probability converging to 1.

Hence, $\|\widehat{D} - D\|_F = \mathcal{O}\left( \dfrac{m}{n_0} \right)$ with probability converging to 1 as $n \to \infty$ and $\dfrac{m^2 n_0^2 \log n}{n} \to 0$.

A variant of Davis-Kahan theorem (Mukherjee et al., 2017) and the derived deviation bound (24) are used to prove the following lemma.

**Lemma 5 (Davis-Kahan theorem)** *Let $V$ and $\widehat{V}$ be the $m \times K$ matrices whose columns correspond to the leading $K$ eigenvectors of $D$ and $\widehat{D}$, respectively. Let $\gamma$ be the $K$-th smallest eigenvalue value of $D$ in magnitude. As $n \to \infty$ and $\dfrac{n_0^2 \log n}{n} \to 0$, there exists an orthogonal matrix $\widehat{O}$ such that,*

$$\left\| \widehat{V}\widehat{O} - V \right\|_F = \mathcal{O}\left( \frac{m}{\gamma n_0} \right) \tag{25}$$

*with probability converging to 1.*

**Proof of Lemma 5 (Davis-Kahan theorem).** A variant of Davis Kahan theorem from Proposition A.2 of Mukherjee et al. (2017) states the following for matrix $D$ of rank $K$. Let $\widehat{V}$ and $V$ be $m * K$ matrices whose columns correspond to the leading $K$ eigenvectors of $\widehat{D}$ and $D$, respectively, and $\gamma$ be the $K$-th smallest eignenvalue of $D$ in magnitude, then there exists an orthogonal matrix $\widehat{O}$ of size $K * K$ such that,

$$\left\|\widehat{V}\widehat{O} - V\right\|_F \leq \frac{4\left\|\widehat{D} - D\right\|_F}{\gamma} \overset{4}{=} \mathcal{O}\left(\frac{m}{\gamma n_0}\right) \quad \text{as } n \to \infty \text{ and } \frac{m^2 n_0^2 \log n}{n} \to 0. \square$$

The number of misclustered graphs is $|\mathcal{M}| \leq 8m_T\|\widehat{V}\widehat{O} - V\|_F^2$ where $m_T$ is the maximum number of graphs generated from a single graphon (Mukherjee et al., 2017). Since $m_T = \mathcal{O}(m)$, $|\mathcal{M}| = \mathcal{O}\left(\frac{m^3}{\gamma^2 n_0^2}\right)$ by substituting (25) in $|\mathcal{M}|$. Hence proving Theorem 1.

**Proof of Theorem 1.** The number of misclustered graphs $|\mathcal{M}| \leq 8m_T\left\|\widehat{V}\widehat{O} - V\right\|_F^2$ from Mukherjee et al. (2017). Thus, we prove the theorem using Lemma 5. That is, as $n \to \infty$ and $\frac{m^2 n_0^2 \log n}{n} \to 0$,

$$|\mathcal{M}| \leq 8m_T\left\|\widehat{V}\widehat{O} - V\right\|_F^2 \overset{5}{=} \mathcal{O}\left(\frac{m^3}{\gamma^2 n_0^2}\right) \square$$

**Proof of Corollary 1.** This corollary deals with a special case where $K = 2$ and equal number of graphs are generated from the two graphons $w$ and $w'$. Therefore, $m_T$ in the number of misclustered graphs $|\mathcal{M}|$ is $m/2$. The ideal distance matrix $D$ will be of size $m \times m$ with 0 and $\|w - w'\|_{L_2}$ as entries depending on whether the samples are generated from the same graphon or not. For such a block matrix $D$, the two non zero eigenvalues are $\pm\frac{m}{2}\|w - w'\|_{L_2}$. Therefore, $\gamma$ is $\frac{m}{2}\|w - w'\|_{L_2}$. Corollary 1 can be derived by substituting the derived $\gamma$ in the number of misclustered graphs $|\mathcal{M}|$ in Theorem 1 as shown below.

$$|\mathcal{M}| = \mathcal{O}\left(\frac{m}{\|w - w'\|_{L_2}^2 n_0^2}\right)$$

Let us assume $\|w - w'\|_{L_2} \geq C\frac{m}{n_0}$ where $C$ is a large constant, then as $n \to \infty$, $\frac{m^2 n_0^2 \log n}{n} \to 0$, $|\mathcal{M}| \to 0$. $\square$

## A.4 Similarity Based Semi-Definite Programming (SSDP)

We make Assumptions 1–3 on the $K$ graphons to study the recovery of clusters from Algorithm 2. The proof structure for cluster recovery stated in Theorem 2 is as follows:

1. We establish deviation bound between the estimated similarity matrix $\widehat{S}$ and the ideal similarity matrix $S$ (Lemma 6).
2. We derive the recoverability condition by adapting Proposition 1 of Perrot et al. (2020) and the obtained deviation bound (Lemma 7).

The ideal similarity matrix $S \in \mathbb{R}^{m \times m}$ is symmetric with $K \times K$ block structure, and $S = Z\Sigma Z^T$ where $Z \in \{0, 1\}^{m \times K}$ be the clustering membership matrix and $\Sigma \in \mathbb{R}^{K \times K}$ such that $\Sigma_{ll'}$ represents ideal pairwise similarity between graphs from clusters $\mathcal{C}_l$ and $\mathcal{C}_{l'}$. From the definition of $S_{ij}$, $\Sigma_{ll'} = \exp\left(-\frac{\|w_l - w_l\|_{L_2}}{\sigma_l \sigma_{l'}}\right)$ where $w_l$ and $w_{l'}$ are graphons corresponding to clusters $\mathcal{C}_l$ and $\mathcal{C}_{l'}$, respectively. $\widehat{S}$ is the estimated similarity matrix of $S$ as mentioned earlier. Since $\widehat{X} \in \mathbb{R}^{K \times K}$ is the normalised clustering matrix, $\widehat{X} = ZN^{-1}Z^T$ where $N$ is a diagonal matrix with $\frac{1}{|\mathcal{C}_1|}, \dots, \frac{1}{|\mathcal{C}_K|}$. We derive the deviation bound for the similarity matrices using Lemma 3 and the result is as follows.

**Lemma 6 (Similarity deviation bound)** *As $n \to \infty$, $\dfrac{n_0^2 \log n}{n} \to 0$, we establish*

$$|\widehat{S}_{ij} - S_{ij}| = \mathcal{O}\left(\frac{1}{n_0}\right) \tag{26}$$

*with probability converging to 1. Hence, from the result $\|\widehat{S} - S\|_F = \mathcal{O}\left(\dfrac{m}{n_0}\right)$ with probability converging to 1.*

**Proof of Lemma 6 (Similarity deviation bound).** We derive the bound using Lemmas 2 and 3, and the definitions of $\widehat{S}_{ij}$ and $S_{ij}$.

$$
\begin{aligned}
\widehat{S}_{ij} &= \exp\left(-\frac{\|A_i - A_j\|_F}{n_0 \sigma_i \sigma_j}\right) && \text{Consider } \sigma_i = \sigma_j = \sigma \\[2mm]
&\overset{3}{\geq} \exp\left(-\frac{\|W_i - W_j\|_F + 4\epsilon}{n_0 \sigma^2}\right) && \text{with probability} \to 1 \text{ asymptotically} \\[2mm]
&\overset{2}{\geq} \exp\left(-\frac{\|w_i - w_j\|_{L_2}}{\sigma^2}\right) \exp\left(-\frac{4\epsilon + 4\sqrt{2}L}{n_0 \sigma^2}\right) && \exp(-x) \geq 1 - 2x \text{ for } x > 0 \\[2mm]
&\geq S_{ij}\left(1 - \frac{8(\epsilon + \sqrt{2}L)}{n_0 \sigma^2}\right) && \\[2mm]
&\geq S_{ij} - S_{ij}\frac{8(\epsilon + \sqrt{2}L)}{n_0 \sigma^2} && S_{ij} \in [0, 1] \\[2mm]
&\geq S_{ij} - \frac{8(\epsilon + \sqrt{2}L)}{n_0 \sigma^2} && 
\end{aligned}
\tag{27}
$$

$$
\begin{aligned}
\widehat{S}_{ij} &= \exp\left(-\frac{\|A_i - A_j\|_F}{n_0 \sigma^2}\right) && \\[2mm]
&\overset{3}{\leq} \exp\left(-\frac{\|W_i - W_j\|_F - 4\epsilon}{n_0 \sigma^2}\right) && \text{with probability} \to 1 \text{ asymptotically} \\[2mm]
&\overset{2}{\leq} \exp\left(-\frac{\|w_i - w_j\|_{L_2}}{\sigma^2}\right) \exp\left(\frac{4\epsilon + 4\sqrt{2}L}{n_0 \sigma^2}\right) && \exp(x) \leq 1 + 2x \text{ for } x > 0 \\[2mm]
&\leq S_{ij}\left(1 + \frac{8(\epsilon + \sqrt{2}L)}{n_0 \sigma^2}\right) && \\[2mm]
&\leq S_{ij} + S_{ij}\frac{8(\epsilon + \sqrt{2}L)}{n_0 \sigma^2} && S_{ij} \in [0, 1] \\[2mm]
&\leq S_{ij} + \frac{8(\epsilon + \sqrt{2}L)}{n_0 \sigma^2} && 
\end{aligned}
\tag{28}
$$

Thus, from (27) and (28), we get $|\widehat{S}_{ij} - S_{ij}| \leq \dfrac{8(\epsilon + \sqrt{2}L)}{n_0 \sigma^2} = \mathcal{O}\left(\dfrac{1}{n_0}\right)$ for any $\epsilon$, with probability converging to 1 as $n \to \infty$ and $\dfrac{n_0^2 \log n}{n} \to 0$. Hence, $\|\widehat{S} - S\|_F \leq \dfrac{8m(\epsilon + \sqrt{2}L)}{n_0 \sigma^2} = \mathcal{O}\left(\dfrac{m}{n_0}\right)$, with probability converging to 1 as $n \to \infty$ and $\dfrac{m^2 n_0^2 \log n}{n} \to 0$. $\qquad\square$

The condition for exact recovery of clusters is derived by adapting Proposition 1 of Perrot et al. (2020). The proposition states the recoverability condition for such an SDP defined in (4) in terms of the similarity deviation bound. Thus, we use the derived bound in Lemma 6 and establish condition on the $L_2$-distance to satisfy the proposition from Perrot et al. (2020). First, we state the adapted proposition.

We define $\Delta_1$ and $\Delta_2$ as,

$$\Delta_1 = \min_{l \neq l'}(1 - \Sigma_{ll'}) \quad \text{and} \quad \Delta_2 = \max_{ij}|\widehat{S}_{ij} - S_{ij}|.$$

Then, the following should be satisfied for $\widehat{X}$ to be the unique optimal solution of the SDP in (4):

$$\|\widehat{S} - S\|_F \leq \min_l |\mathcal{C}_l| \min\left\{\frac{\Delta_1}{2}, \Delta_1 - 6\Delta_2\right\}.$$

The minimum cluster size $\min_l |\mathcal{C}_l|$ in our case is 1. Consequently, the recoverability condition is derived and is as follows.

**Lemma 7 (Recoverability of clusters)** *As $n \to \infty$, $\dfrac{m^2 n_0^2 \log n}{n} \to 0$, the $\min_{l \neq l'}\|w_l - w_l'\|_{L_2}$ should be $\Omega\left(\dfrac{m}{n_0}\right)$ so that $\widehat{X}$ is the unique optimal solution of the SDP (4).*

**Proof of Lemma 7 (Recoverability of clusters).** We derive the condition to satisfy the stated proposition.

$\Delta_1 = 1 - \max_{l \neq l'} \Sigma_{ll'}$ and $\Delta_2 = \min_{ij}|\widehat{S}_{ij} - S_{ij}|$. The minimum cluster size in our case can be 1.

The analyses of the two cases of the Proposition is as follows.

**Case 1.** Let us assume $\Delta_2 \leq \dfrac{\Delta_1}{12}$, then $\min\left\{\frac{\Delta_1}{2}, \Delta_1 - 6\Delta_2\right\}$ will be $\frac{\Delta_1}{2}$. Therefore,

$$\frac{8m(\epsilon + \sqrt{2}L)}{n_0\sigma^2} \leq \frac{1}{2} - \frac{1}{2}\max_{l \neq l'}\exp\left(-\frac{\|w_l - w_{l'}\|_{L_2}}{\sigma^2}\right)$$

$$\exp\left(-\min_{l \neq l'}\frac{\|w_l - w_{l'}\|_{L_2}}{\sigma^2}\right) \leq 1 - \frac{16m(\epsilon + \sqrt{2}L)}{n_0\sigma^2}$$

$$\min_{l \neq l'}\frac{\|w_l - w_{l'}\|_{L_2}}{\sigma^2} \geq -\log\left(1 - \frac{16m(\epsilon + \sqrt{2}L)}{n_0\sigma^2}\right)$$

$$\min_{l \neq l'}\|w_l - w_{l'}\|_{L_2} \geq \sigma^2 \sum_{k=1}^{\infty}\left(\frac{16m(\epsilon + \sqrt{2}L)}{n_0\sigma^2}\right)^k \frac{1}{k}$$

$$\min_{l \neq l'}\|w_l - w_{l'}\|_{L_2} = \Omega\left(\frac{m}{n_0}\right) \tag{29}$$

**Case 2.** Let us assume $\Delta_2 > \dfrac{\Delta_1}{12}$, then $\min\left\{\frac{\Delta_1}{2}, \Delta_1 - 6\Delta_2\right\}$ will be $\Delta_1 - 6\Delta_2$. Therefore,

$$\frac{8m(\epsilon + \sqrt{2}L)}{n_0\sigma^2} \leq 1 - \max_{l \neq l'}\exp\left(-\frac{\|w_l - w_{l'}\|_{L_2}}{\sigma^2}\right) - \frac{6 * 8(\epsilon + \sqrt{2}L)}{n_0\sigma^2} \text{ with probability} \to 1 \text{ aymptotically}$$

$$\min_{l \neq l'}\frac{\|w_l - w_{l'}\|_{L_2}}{\sigma^2} \geq -\log\left(1 - \frac{8(m + 6)(\epsilon + \sqrt{2}L)}{n_0\sigma^2}\right)$$

$$\min_{l \neq l'}\|w_l - w_{l'}\|_{L_2} \geq \sigma^2 \sum_{k=1}^{\infty}\left(\frac{8(m + 6)(\epsilon + \sqrt{2}L)}{n_0\sigma^2}\right)^k \frac{1}{k}$$

$$\min_{l \neq l'}\|w_l - w_{l'}\|_{L_2} = \Omega\left(\frac{m}{n_0}\right) \tag{30}$$

Thus, from (29) and (30), we must satisfy $\min_{l \neq l'}\|w_l - w_{l'}\|_{L_2} = \Omega\left(\dfrac{m}{n_0}\right)$ for the Proposition to hold. Consequently, Theorem 2 is the direct reflection of this lemma. $\square$

A.5   GRAPH TWO-SAMPLE TESTING

Theorem 3 of two-sample testing is proved by deriving the probability of Type-1 and Type-2 errors. We make Assumptions 1–3 for this case. Let $W_1$ and $W_2$ be the $n_0 \times n_0$ discretized graphons of $w_1$ and $w_2$, respectively, obtained using (6). Then, the alternate hypothesis $H_a$ can be rewritten using Lemma 2 in the following way,

$$\frac{1}{n_0} \|W_1 - W_2\|_F + \frac{4\sqrt{2}L}{n_0} \overset{(9)}{\geq} \phi$$

$$\|W_1 - W_2\|_F \geq n_0\phi - 4\sqrt{2}L = \rho \tag{31}$$

We derive the probability of the errors using Lemma 3 and is stated in the following lemmas.

**Lemma 8 (Probability of Type-1 error)** *The probability of Type-1 error, i.e. rejecting the null hypothesis when it is actually true, is*

$$\mathbb{P}(T = 1 | H_0 : True) \leq \frac{C}{\xi^2} \frac{\log n}{n} \tag{32}$$

*where $C$ depends only on the Lipschitz constants.*

**Proof of Lemma 8 (Probability of Type-1 error).** The Type-1 error is rejecting $H_0$ when it is true. Therefore, in this scenario, $\|W_1 - W_2\|_F = 0$. Thus, from Lemma 3, (15) and (17), we have $\|A_1 - A_2\|_F \leq 4\epsilon$ with $1 - \dfrac{C}{\epsilon^2}\dfrac{n_0^2 \log n}{n}$ probability. Therefore, the probability of Type-1 error is,

$$\begin{aligned}
\mathbb{P}(T = 1 | H_0 : True) &= \mathbb{P}(d(G_1, G_2) \geq \xi) \\
&= \mathbb{P}(\|A_1 - A_2\|_F \geq n_0\xi) \qquad \text{err only when } n_0\xi \leq 4\epsilon \\
&\overset{(15)}{\leq} \frac{C}{\xi^2 n_0^2} \frac{n_0^2 \log n}{n} \square
\end{aligned}$$

**Lemma 9 (Probability of Type-2 error)** *The probability of Type-2 error, i.e. accepting the null hypothesis when the alternate hypothesis is actually true, is*

$$\mathbb{P}(T = 0 | H_a : True) \leq \frac{C_1}{\left(\phi - \frac{4\sqrt{2}C_2}{n_0}\right)^2} \frac{\log n}{n} \tag{33}$$

*where $C_1$ and $C_2$ depend only on the Lipschitz constants.*

**Proof of Lemma 9 (Probability of Type-2 error).** The Type-2 error is evaluating to null hypothesis when the alternate hypothesis is true. Therefore, from (31) $\|W_1 - W_2\|_F \geq \rho$. From Lemma 3, (15) and (17),

$$\|A_1 - A_2\|_F \geq \|W_1 - W_2\|_F - 4\epsilon \qquad \text{with probability } 1 - \frac{C_1}{\epsilon^2} \frac{n_0^2 \log n}{n}$$

$$\geq \rho - 4\epsilon$$

The probability of Type-2 error is,

$$\begin{aligned}
\mathbb{P}(T = 0 | H_a : True) &= \mathbb{P}(d(G_1, G_2) < \xi) \\
&= \mathbb{P}(\|A_1 - A_2\|_F < n_0\xi) \quad \text{err only when } n_0\xi \leq \rho - 4\epsilon; \text{ let } n_0\xi = 4\epsilon \\
&\leq \mathbb{P}(\|A_1 - A_2\|_F < \frac{\rho}{2}) \\
&\leq \frac{C_1}{\rho^2} \frac{n_0^2 \log n}{n}
\end{aligned}$$

We get the probability by substituting $\dfrac{\rho}{n_0} = \phi - \dfrac{4\sqrt{2}L}{n_0}$ from (31) in the above equation. Theorem 3 can be proved by asymptotic analysis of Lemmas 8 and 9. $\square$

## B   DSC AND SSDP ALGORITHMS

The proposed algorithms DSC and SSDP are described as follows:

| **Algorithm 1:** Distance based Spectral Clustering (DSC) | **Algorithm 2:** Similarity based Spectral Clustering (SSDP) |
|---|---|
| **input** : Adjacency matrices $G_1, ..., G_m$, histogram size $n_0$ 
 **output:** $K$ clusters $\mathcal{C}_1, ..., \mathcal{C}_K$ | **input** : Adjacency matrices $G_1, ..., G_m$, histogram size $n_0$ 
 **output:** $K$ clusters $\mathcal{C}_1, ..., \mathcal{C}_K$ |
| **Construct distance matrix** Compute $\widehat{D} \in \mathbb{R}^{m \times m}$, where $\widehat{D}_{ij} = d(G_i, G_j)$ | **Construct similarity matrix** Compute $\widehat{S} \in \mathbb{R}^{m \times m}$, where $\widehat{S}_{ij} = \exp\left(-\frac{d(G_i, G_j)}{\sigma_i \sigma_j}\right)$ with $\sigma_1 = \ldots = \sigma_n$ |
| **Clustering** Apply spectral clustering to $\widehat{D}$ with $K$ number of clusters resulting in $\mathcal{C}_1, ..., \mathcal{C}_K$ | **Clustering** Find $\widehat{X}$ using (4) and apply standard spectral clustering to $\widehat{X}$ resulting in $\mathcal{C}_1, ..., \mathcal{C}_K$. |

## C   EXPERIMENTAL DETAILS

In this section, we present experimental details and additional experiments.

### C.1   SIMULATED DATA - HEATMAP OF GRAPHONS

Figure 4 shows the heatmap of the considered four graphons $W_1, W_2, W_3$ and $W_4$. We sample graphs from these graphons for the experiments.

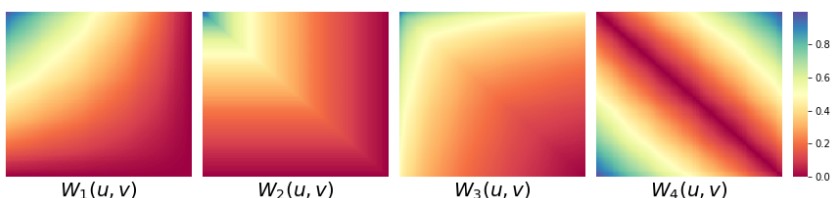

Figure 4: Heatmaps of graphons $W_1, W_2, W_3$ and $W_4$.

### C.2   CHOICE OF $n_0$

We validate the theoretically deduced bound for $n_0 = \mathcal{O}(\sqrt{n/\log n})$ by sampling 5 graphs with a fixed number of nodes $n$ from each of the four graphons, in total 20 graphs, and measuring the performance of DSC and SSDP for different $n_0 = \{5, 10, 15, 20, 25, 30\}$. We perform three simulations with $n = \{50, 100, 500\}$ and fix neighbourhood of one in SSDP. Figure 5 shows the average error of both the algorithms over 5 independent trials. Based on the theoretical considerations for $n_0$ ($\ll \sqrt{n/\log n}$), we evaluate $n_0 = \{5, 7, 15\}$ for $n = \{50, 100, 500\}$, respectively. The experimental results show that the derived bound for $n_0$ serves as a reasonable choice (if not the best) for both DSC and SSDP irrespective of $n$. Hence, the choice of $n_0$ can be deterministic and adaptive with respect to $n$, thus making our algorithms parameter-free.

### C.3   EXPERIMENTAL RESULTS USING ADJUSTED RAND INDEX (ARI)

In this section, we provide the results for evaluation of algorithms on simulated and real data under the same setting as described in Section 3.3. Figure 6 shows the evaluation of all the discussed algorithms using ARI, where the observations made from error rate hold.

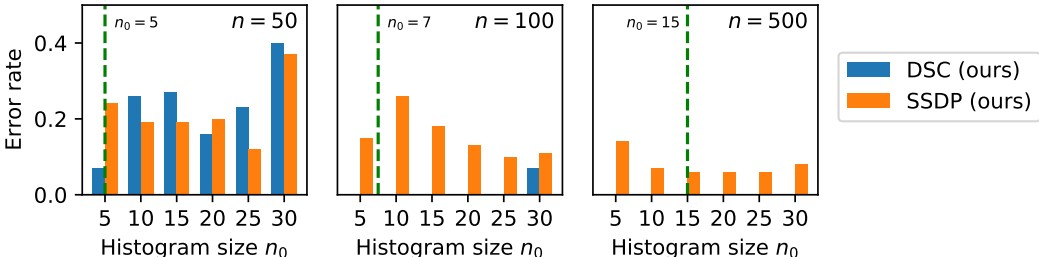

Figure 5: Validation of the bound for $n_0$. The plot shows the average error rate (percentage of misclustered graphs) of the proposed algorithms DSC and SSDP for different $n = \{50, 100, 500\}$.

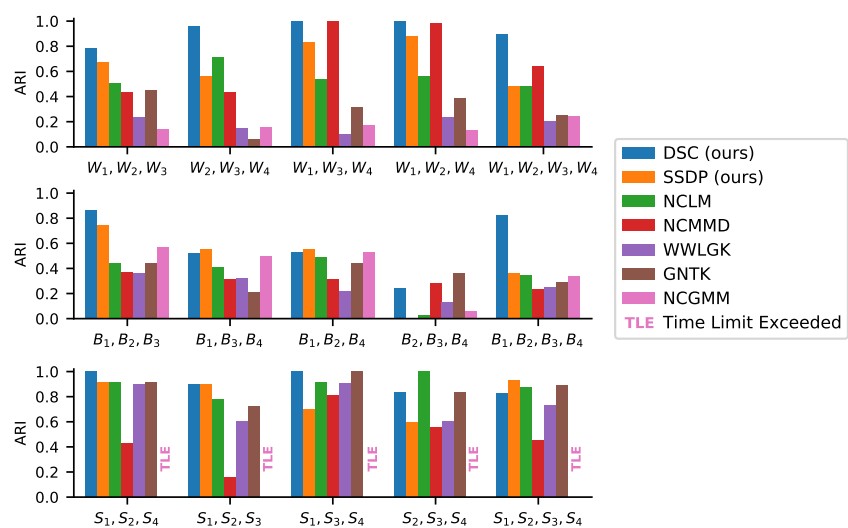

Figure 6: Evaluation of DSC and SSDP with other methods. **(row 1)** Results on simulated data using ARI. **(rows 2 and 3)** Results on real data from Bioinformatics and Social Networks, respectively.

## C.4 EVALUATION ON LARGE SIMULATED DATA

As mentioned in Section 3.3, we evaluate algorithms except GNTK and NCGMM on large graphs sampled from the four graphons $W_1, W_2, W_3$ and $W_4$ with nodes between $100$ and $1000$. Figure 7 shows the results measured using average error rate and average ARI, respectively. The proposed algorithm DSC outperforms the others in all the case and SSDP stands second or third best, as observed in simulated data with small graphs.

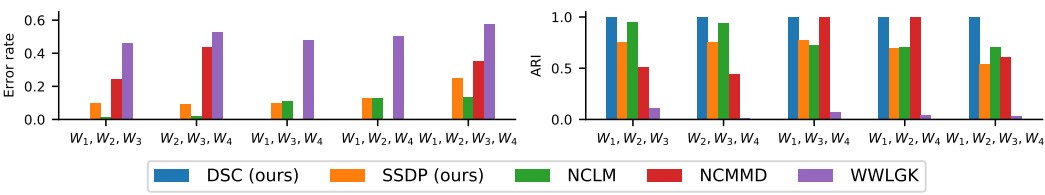

Figure 7: Evaluation of all algorithms except GNTK and NCGMM using average error rate and average ARI for large simulated data.

## C.5 COMPUTATION TIME OF ALGORITHMS

Table 1 shows the time (measured in seconds) taken for the algorithms on the considered dataset combinations. Three clusters also show similar behavior to four clusters.

Table 1: Time measured in seconds for algorithms on different datasets combinations

| Dataset | DSC | SSDP | NCLM | NCMMD | WWLGK | GNTK | NCGMP |
|---|---|---|---|---|---|---|---|
| $W_1, W_2, W_3$ | 0.13 | 0.22 | 0.17 | 11.66 | 0.66 | 225.46 | 15.43 |
| $W_2, W_3, W_4$ | 0.14 | 0.23 | 0.18 | 12.98 | 0.70 | 199.98 | 16.27 |
| $W_1, W_3, W_4$ | 0.13 | 0.25 | 0.19 | 12.93 | 0.71 | 200.85 | 16.98 |
| $W_1, W_2, W_4$ | 0.13 | 0.24 | 0.18 | 12.55 | 0.68 | 198.38 | 16.54 |
| $W_1, W_2, W_3, W_4$ | 0.20 | 0.38 | 0.28 | 22.05 | 1.19 | 390.07 | 30.18 |
| $B_1, B_2, B_3$ | 1.10 | 1.25 | 0.17 | 20.72 | 2.54 | 28.21 | 219.58 |
| $B_1, B_3, B_4$ | 0.99 | 1.18 | 0.16 | 21.51 | 2.85 | 32.22 | 225.02 |
| $B_1, B_2, B_4$ | 1.01 | 1.08 | 0.14 | 15.61 | 1.92 | 19.49 | 174.25 |
| $B_2, B_3, B_4$ | 1.04 | 1.13 | 0.13 | 11.99 | 0.78 | 13.38 | 21.47 |
| $B_1, B_2, B_3, B_4$ | 1.33 | 1.46 | 0.183 | 28.66 | 3.19 | 40.18 | 278.80 |
| $S_1, S_2, S_4$ | 1.50 | 1.65 | 8.21 | 1125.71 | 454.19 | 1609.78 | TLE |
| $S_1, S_2, S_3$ | 1.25 | 1.34 | 0.36 | 77.67 | 15.32 | 294.53 | TLE |
| $S_1, S_3, S_4$ | 1.52 | 1.64 | 8.07 | 1001.52 | 348.25 | 1485.90 | TLE |
| $S_2, S_3, S_4$ | 1.48 | 1.69 | 8.88 | 1035.98 | 440.87 | 1757.06 | TLE |
| $S_1, S_2, S_3, S_4$ | 1.97 | 2.22 | 9.47 | 1069.28 | 437.21 | 2060.68 | TLE |

## C.6 SCALABILITY EXPERIMENT

We evaluate the scalability of the considered algorithms using simulated data by measuring the time taken for clustering 40 random graphs, 10 sampled from each of the graphons $W_1, W_2, W_3$ and $W_4$. We did 7 experiments in which the size of the sampled graphs are varied as $[50, \text{max\_size}]$ where max\_size $= \{100, 200, 300, 400, 500, 600, 700\}$. Figure 8 shows the experimental results which illustrates high scalability of DSC, SSDP and NCLM over other algorithms. Note that the experiment shows NCLM as scalable as DSC and SSDP since the sampled graphs are small.

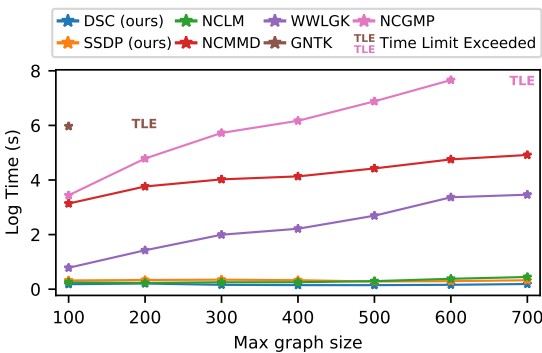

Figure 8: Computation time of algorithms on different sets of simulated data for four clusters case demonstrating the scalability of each algorithm. Computation time is plotted in log scale.

## C.7 TWO-SAMPLE TESTING

In this section, we evaluate the efficacy of the proposed test $T$ with different $d(G_1, G_2)$ by varying the graph sizes $n$. We consider $n = \{50, 100, 150\}$ and fix $n_0 = 10$ from the theoretical bound for evaluating the test $T$. The power is computed using the test $T$ for the significance level $0.05$, and the plots in Figure 9 show the average power computed over 500 trials of bootstrapping 100 samples generated from all pairs of graphons for $d(G_1, G_2)$ as our proposed distance, log moments

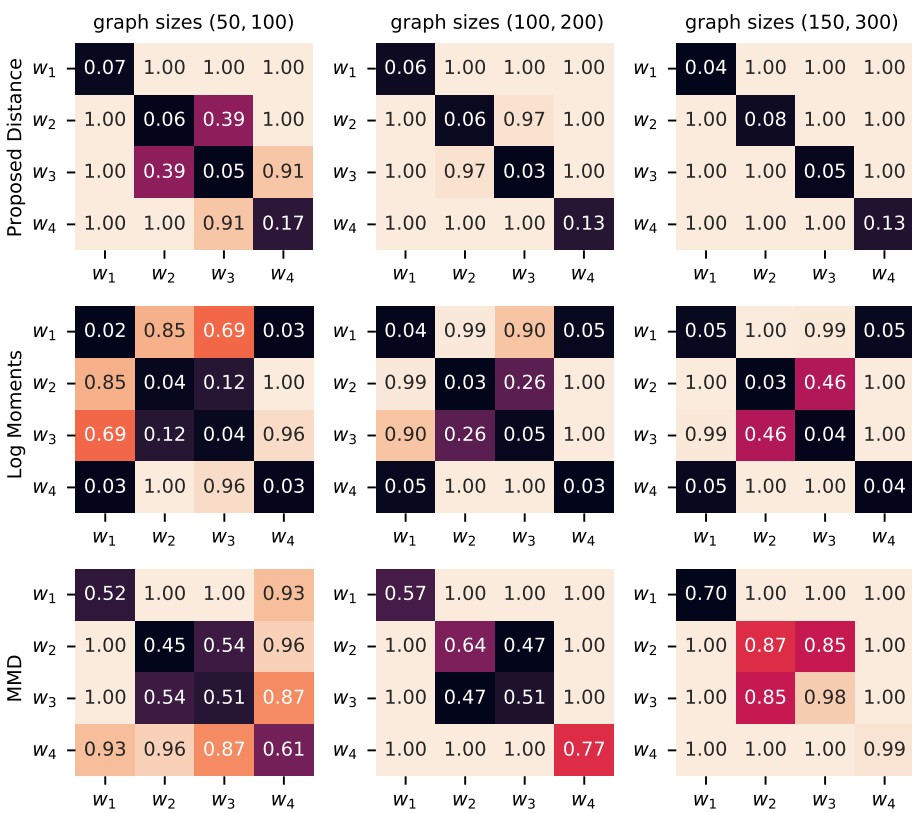

Figure 9: Illustration of two-sample testing with the proposed distance vs log moments and MMD on varying $n$ in graph pairs of size $(n, 2n)$. The plots show the average power of the test $T$. Test based on the proposed distance is consistent for sufficiently large graphs and efficient compared to other methods in distinguishing even closer graphons.

and MMD, respectively. From the result for graph sizes $(50, 100)$, we observe that the graphon pair $(W_2, W_3)$ is not easily distinguishable (low $H_0$ rejection probability), which can be explained by their respective $L_2$ distance that is shown in the left plot of Figure 3. This issue does not arise in testing larger graphs as the result shows for graph sizes $(100, 200)$ and $(150, 300)$. Therefore, test $T$ with the proposed distance can distinguish between pairs of graphons that are quite close provided that the observed graphs are sufficiently large, thus proving to be consistent. On the other hand, log moments and MMD based tests show weakness in distinguishing the graphons, where log moments based test $T$ accepts the null hypothesis in most cases even when the graphons are different for all graph sizes. For instance, the result for graphon pair $W_1$ and $W_4$ is indistinguishable using log moments statistic for any graph size. On the contrary, MMD based test $T$ rejects the null hypothesis almost always for larger graphs (diagonal values in all graph cases). Thus, we conclude that the proposed test $T$ in 5 is consistent and this experiment illustrates the efficiency of the test $T$ compared to other plausible test statistics.

Subsequently, we evaluate the efficacy of the above tests on the discussed real datasets – Bioinformatics and Social Networks. We consider graphs from a dataset to belong to a population and hence the objective of the test statistic is to distinguish graphs from different populations, that is, graphs from two different datasets. Since the populations are not known and the real graphs are treated as representatives of the population, we compute $p$-value of the test instead of power to measure the efficacy. The $p$-value is the evidence for rejecting the null hypothesis which implies that the smaller the $p$-value, the stronger the evidence that the null hypothesis should be rejected. Therefore, the $p$-value should be high (greater than the significance level) for graphs from same population and low ($\simeq 0$) for graphs from different populations. Figure 10 shows the result for both the dataset cases and different tests. From the results, it is clear that the log moments and MMD based tests are poor

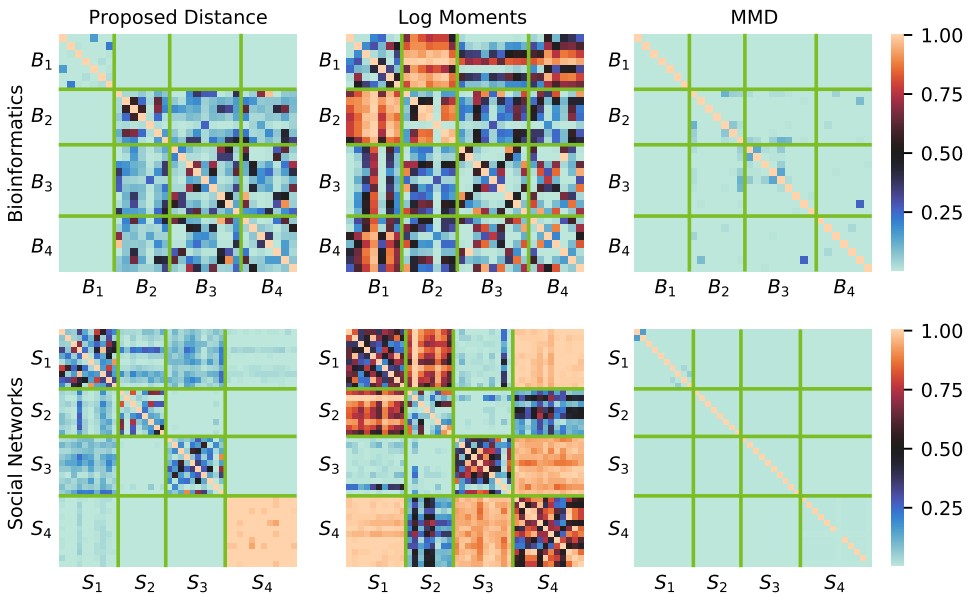

Figure 10: Illustration of two-sample testing with the proposed distance vs log moments and MMD on real datasets - Bioinformatics and Social Networks. The plots show the $p$-value of the test $T$. Test based on the proposed distance is better than the other two tests.

and inefficient on real datasets as log moments based test has high acceptance of null hypothesis for almost all the pair of graphs from any population and MMD based test rejects the null hypothesis always except when the graphs are the same. Whereas, the test using our proposed distance perform well on large graphs from Social Networks datasets, for instance, $S_4$ with other datasets and within itself. This test performs well even for small graphs when compared to the other two tests.

