# OpenReview forum: "Graphon based Clustering and Testing of Networks: Algorithms and Theory"
_ICLR.cc/2022/Conference — ICLR 2022 Poster_

### Official Review · Reviewer_QUvS · 2021-11-01

**Correctness:** 3
**Technical Novelty And Significance:** 4
**Empirical Novelty And Significance:** 3
**Recommendation:** 8
**Confidence:** 3

**Main Review:**


The paper is very interesting, well written and understandable.

The proposed heory only works when graphs are sampled from graphons and the vertices are kept in the same order.  On the one hand, I understand that anything involving graph isomrophisms (and hence vertex permutations) will be untractable.  On the other hand, in practical applications we don't know the order of the vertices, hence there is a lot of work on graph isomorphism, graphon cut norms, and other tools to deal with that issue.  Assuming instead 'ordered' graph sampling with L2 graphon norms is interesting from a theory point of view but is quite far from practical applications where clustering and testing becomes relevant.

The experimental section could devote more attention to the runtime of the algorithms and experiments.

Nevertheless, the paper has some original ideas which could form a first step into more mature results.


Some details:
* Assumption 2: I dislike the term "degree distribution" in this context.  Usually, "degree distribution" is used for a probability distribution f which gives for a input degree d the probability f(d) that a randomly selected vertex has degree d.

* In Equation (1), it seems the index i of A_{i,j} starts from 0 until n_0-1 rather than from 1 to n_0 as usual.  Alos, vertices with index larger than n_0*\lfloor n/n_0 \rfloor aren't taken into account.

at some places articles are missing, e.g.,
* page 5 first line: Davis-Kuhan theorem -> the Davis-Kuhan theorem
* Sec 3.2 second line: Gaussian kernel -> the Gaussian kernel





**Summary Of The Paper:**


This paper presents an interesting approach to clustering of graphs sampled from graphons.  The papers presents a number of guarantees for the quality of obtained clusterings and a few preliminary experiments.


**Summary Of The Review:**



The paper is interesting and sound, while the result on itself is not yet very practical it could form a good first step.

---

> ### Author Response · Authors · 2021-11-14
> **Response to reviewer's comments (runtime of algorithms, practicality of the methods)**
>
> We thank the reviewer for the positive feedback and appreciation for the theory in our work. The technical details and minor comments are incorporated in the paper.
>
> ### 1. Runtime of algorithms in experiments
> We refer the reviewer to the detailed discussion on the runtime of algorithms in Section 3.3 under ‘Computation time comparison’ and Table 1 in Appendix C.5 has the exact time taken by the algorithms on all the considered datasets combinations. We would be happy to address if the reviewer points out the missing aspects of runtime.
>
> ### 2. Practicality of the methods
> We are not sure about the reviewer's comment on the practicality of our methods, and we would be pleased to have a discussion on this.
> We agree with the reviewer that our theoretical framework is impractical since it only considers dense graphs (also see Response 1 to comments on sparse graphs by Reviewer fKkm) . Despite the consistency results established primarily on restrictive assumptions, our algorithms proved to be powerful (both in terms of accuracy and computation) on a broad category of real datasets ranging from small (Bioinformatics) to large graphs (Social Networks). Hence, we think the proposed algorithms are practically feasible and competitive, regardless of the fact that the theory can certainly be improved.

---

### Official Review · Reviewer_1avZ · 2021-11-02

**Correctness:** 4
**Technical Novelty And Significance:** 3
**Empirical Novelty And Significance:** Not applicable
**Recommendation:** 6
**Confidence:** 4

**Main Review:**

__Strengths__:

The paper is easy to read and follow, well-grounded, and theoretically sound in most of its parts.
The authors also try to give intuition about the choices they made.
The developed method is applicable and effective to small samples of graphs and it appears novel to me.

__Weaknesses__:
1. The authors claim there are no studies beyond network classification and a lack of theoretical support for the available methods. In my opinion, the claim is not correct. Many papers include different graph-level tasks in their experiments and several of them provide also theoretical support about the proposed architecture; I can refer the authors to some of them below. The authors also claim "the efficacy of these methods [on graph matching] in learning from network-valued data remains unexplored". The authors should be more precise about what remains unexplored. Nevertheless, I may agree that extra studies and theoretical analyses can positively contribute to the research in this field, and I recognize the value of considering the small sample size setting.
    * https://ogb.stanford.edu/
    * Dwivedi, Vijay Prakash, et al. 2020. Benchmarking graph neural networks
    * Bouritsas, Giorgos, et al. 2020. Improving graph neural network expressivity via subgraph isomorphism counting.
    * Zambon, D., Alippi, C., & Livi, L. 2020. Graph Random Neural Features for Distance-Preserving Graph Representations.
    * Sato, Ryoma. 2020. A survey on the expressive power of graph neural networks.
    * Maron, Haggai, Heli Ben-Hamu, and Yaron Lipman. 2019. Open problems: Approximation power of invariant graph networks.
    * Loukas, Andreas. 2019. What graph neural networks cannot learn: depth vs width.
    * Kriege, Nils M., et al. 2018. A Property Testing Framework for the Theoretical Expressivity of Graph Kernels.
    * Chen, Hao, and Jerome H. Friedman. 2017. A new graph-based two-sample test for multivariate and object data.
2. The graph transformation of G to G^\sigma seems to be ill-defined. Although the graphon can have unique node degrees, graphs sampled from it can result in nodes with the same degree, so 1) there might not exist a monotonically _increasing_ permutation but only _non-decreasing_, and 2) the permutation is not unique, in general.
3. The claim "NCLM is the only known clustering strategy for graphs of different sizes" is incorrect. Basically, any kernel-based and distance-based clustering method that does not require explicit vector embeddings can be applied to graphs. It is customary to choose the most appropriate distance or kernel that better fits the problem at hand; a simple example is k-means. The claims seem unjustified to me also because, as far as I can tell, NCLM constructs vector representations of graphs.
4. The methods chosen for the comparison can give only limited insights. The considered methods appear to be more or less arbitrarily chosen combinations of graph functions with clustering methods. For example, this approach does not allow us to understand whether it is the proposed distance or the considering clustering method that brings the most advantages. I might be wrong, but it seems to me that there is no limitation in performing experiments with the proposed graphon-based distance with other clustering methods, and the considered clustering methods with other distances/kernels.
5. Regarding the two-sample hypothesis test, it is unclear how a critical region of a given significance level can be constructed without having knowledge about the underlying graphons. From what I see, in the paper, the critical region is computed by sampling from the original graphons. I would to have some clarifications in this respect.


Questions and unclear parts:
- The sorting-and-smoothing estimator is mentioned even in the abstract, but never introduced in the paper. Why is it the only one that can meet the above requirements? Moreover, the above requirements (I guess the authors refer to Assumptions 1-3) are about the graphons, not their estimators.
- Theorem 1. It is counterintuitive for me that for a bounded number of nodes, the more graphs we have, the larger the _rate_ of erroneous clustered graph increases. I would expect that for m -> infinity, the rate of misclustered graphs converges to the optimal value (which is not necessarily zero), and the variance of such observed rate decreases. Could you comment on that?
- It is not clear to me to what extent the discussed clustering methods had to be adapted to work with the proposed distance. It is also unclear whether or not these clustering methods would be consistent regardless of the specific graph distance, provided that the graph distance is sufficiently expressive (eg, it is metric).
- Because the considered method is based on edge histograms, I wonder if there are known limitations or important differences when considering sparse graphs instead of denser ones. Are there different best practices to operate in the two regimes?

__Other comments and suggestions__:
- Say explicitly that graphs are symmetric and without node or edge attributes (if this is the case).
- Expand the comment about indistinguishable inhomogeneous random graph models. What is the argument to show that although different, they are indistinguishable? What are we losing by ignoring such graphs, in what cases are we interested in considering also them?
- Prop 1. Please, expand the discussion about the variable n which seems to be undefined.
- Cor 1. Shouldn't it be: given m, n_0 and |w-w'|, we need to find a _small_ enough constant C?


**Summary Of The Paper:**

The paper proposes a graph distance based on graphons that can operate on the small sample size regime; this is possible by exploiting a large number of nodes.
The proposed paper is meant to address two shortcomings that the authors have found in the literature:
- a lack of study of current graph processing methods beyond graph-level classification, and
- a lack of theoretically sound methods.

The paper also shows theoretical guarantees associated with two existing clustering methods and a two-sample statistical tests when operating with the proposed distance.



**Summary Of The Review:**

Overall the paper is well written. The proposed solutions are novel to me and address an extremely challenging problem related to the small sample size regime. However, a few negative aspects are present: there seem to be incorrect claims, the graph transformation appears not well-defined, and the experimental setup does not bring decisive conclusions.
Given the points raised in the main review section and unless it turns out I misunderstood those parts of the paper, I am afraid to say that the paper is ready for publication in its current state. However, I encourage the authors to carry on the research in this direction, because it appears promising and only few (yet, important) aspects need to be improved.

---

> ### Author Response · Authors · 2021-11-14
> **Clarification on the reviewer's misunderstandings**
>
> Thank you for the review. There appears to be some misunderstandings about the paper which we clarify first before addressing the concerns.
>
> ## Clarification on the misunderstandings
> ### 1. Focus of the paper
> Our primary focus is to develop theoretically consistent algorithms to **cluster graphs of different sizes**, rather than addressing the lack of graph processing methods beyond graph classification. Certainly, we acknowledge that there are several theoretically sound works on graphs and graph based learning problems like classification. To the best of our knowledge, there are no statistically consistent clustering methods that deal with graphs of **different sizes**, which is our interest. There is a difference in the theoretical analysis in the cited papers by the reviewer and the one presented in our paper. We would be glad if the reviewer points out the existing theoretical result that deals with this specific problem.
> ### 2. Novelty of algorithms
> The construction of distance (similarity) matrix $\widehat{D}$ ($\widehat{S}$) using the graph distance is an integral step in the proposed clustering algorithms, DSC (SSDP). Thus, the matrix construction followed by spectral or SDP based clustering should be treated together as the algorithm, which implies DSC and SSDP in the paper are novel and not existing.
> ### 3. Proposed distance vs clustering methods
> The graph distance operates on **two** graphs of any size, which means that the number of samples does not influence it. Whereas, the clustering problem is primarily focused on small sample regime (i.e. clustering a small number of graphs), whose practical use case is justified in the introduction. In addition, we restate that our proposed distance is the basis and integral step in the clustering algorithms, which implies that one cannot compare them.
> ### 4. Choice of different methods and experimental results
> The choice of other clustering methods is not arbitrary as pointed out by the reviewer, rather we chose relevant effective techniques in a range of possible clustering approaches such as embedding based, kernel based and neural network based methods as justified in the introduction. Since we considered all possible approaches to solve clustering different sized graphs and chose the best method(s) in each approach, we argue that the experiments are conclusive.
> ### 5. The reviewer pointed out ambiguity in two sentences
> a. “the efficacy of these methods in learning from network-valued data remains unexplored” - remains unexplored in clustering network-valued data. It is clarified in the paper.
>
> b. “NCLM is the only known clustering strategy for graphs of different sizes“ - we clarify that this method is the only published work that studied *clustering* **different** sized graphs. Undoubtedly one can apply kernel-based or distance-based clustering approaches, but to the best of our knowledge, no work other than NCLM studied this particular problem. Moreover, the efficacy of possible kernel-based and distance-based clustering methods is precisely what we have explored and compared it to the proposed algorithms in the experimental section. We have also rephrased the sentence in the paper.

---

> > ### Author Response · Authors · 2021-11-14
> > **Response to reviewer's concerns/comments**
> >
> > ### 1. Definition of $G^\sigma$
> > We thank the reviewer for pointing this out. $G^\sigma$ is not expected to have strict monotonically increasing degree as well as unique order, since neither the theory nor the empirical observation relies on the unique order. We clarified it in the paper.
> >
> > ### 2. Choice of significance level in two-sample hypothesis test
> > The theory is established for the existence of some $\xi$ that decides the critical region which is a common statistical approach (Tang et al., 2017, Agterberg et al., 2020, Ghoshdastidar et al., 2020). However, in practice, it is often determined by bootstrapping as can be seen in other works such as Tang et al., 2017, Agterberg et al., 2020.
> >
> > ### 3. Graph transformation based on sorting-and-smoothing graphon estimator
> > We require only a particular step of the graphon estimator that deals with transforming the graph and the corresponding discussion can be found under 'Graph transformation' heading in page 3.
> > By ‘the above requirements’, we meant the transformation that maps all graphs into a common metric space and not the assumptions.
> >
> > ### 4. $m \rightarrow \infty$ in Theorem 1
> > We refer the reviewer to the first paragraph in Page 5 after Proof Sketch of Theorem 1, where we have clarified this in detail.
> > Suppose there are 2 graphons and an arbitrarily large number of graphs are sampled from each, but each graph has 10 nodes. In such a case, due to the small sizes of graphs, they can be statistically quite different from the underlying graphon. Hence, one cannot consistently recover the underlying clusters in this case even if $m\to\infty$.
> >
> > ### 5. Sparse vs Dense graphs
> > We refer the reviewer to response 1 to Reviewer fKkm for the discussion on sparse graphs setup.
> > There are different works (Tang et al., 2017, Agterberg et al. 2020 as pointed out by Reviewer fKkm) that study problems on more realistic sparse graphs sampled from graphons by introducing expected fraction of edges in sampling the graph.
> >
> > ### 6. Indistinguishable inhomogeneous random graph models
> > We refer the reviewer to the clarification on Assumption 3 in point 2 to Reviewer fKkm for non-identifiability issue in graphons and why we consider equivalence class in the context of clustering.
> >
> > ### 7. Proposition 1 and Corollary 1
> > * **Proposition 1:** $n$ is undefined - It is stated in the second line of the proposition that the random graphs $G_1$ and $G_2$ with at least $n$ nodes are sampled from the graphons $w_1$ and $w_2$, respectively.
> > * **Corollary 1:** We are not interested in finding the constant $C$. It is some constant for which $\Vert w - w^\prime \Vert_{L_2} \geq C\frac{m}{n_0}$. This form of result is common in learning theory (Chan and Airoldi, 2014, Mukherjee et al., 2017). We would be happy to provide further clarifications if the reviewer has more questions.
> >
> > References:
> > * Stanley Chan and Edoardo Airoldi.  A consistent histogram estimator for exchangeable graph models. In International Conference on Machine Learning, pp. 208–216, 2014.
> > * Soumendu Sundar Mukherjee, Purnamrita Sarkar, and Lizhen Lin.  On clustering network-valued data. In Advances in neural information processing systems, pp. 7071–7081, 2017.
> > * Minh Tang, Avanti Athreya, Daniel L Sussman, Vince Lyzinski, Youngser Park, and Carey E Priebe. A semiparametric two-sample hypothesis testing problem for random graphs. Journal of Computational and Graphical Statistics, 26(2):344–354, 2017a
> > * Joshua Agterberg, Minh Tang, and Carey Priebe.  Nonparametric two-sample hypothesis testing for random graphs with negative and repeated eigenvalues. arXiv preprint arXiv:2012.09828, 2020.
> > * Debarghya Ghoshdastidar, Maurilio Gutzeit, Alexandra Carpentier, Ulrike von Luxburg, et al. Two- sample hypothesis testing for inhomogeneous random graphs. Annals of Statistics, 48(4):2208– 2229, 2020.

---

> > > ### Comment · Reviewer_1avZ · 2021-11-20
> > > **Concerns resolved**
> > >
> > > I thank the authors for their answers and clarifications within the paper. My major concerns appear to be resolved now. I update my score.

---

### Official Review · Reviewer_fKkm · 2021-11-02

**Correctness:** 3
**Technical Novelty And Significance:** 3
**Empirical Novelty And Significance:** 2
**Recommendation:** 8
**Confidence:** 3

**Main Review:**

Strengths:
- Deals with the issue of vertex correspondence for clustering of a population of graphs. A large proportion of existing methods developed for clustering assume vertex matched collection of graphs. Thus the idea is to first employ graph matching algorithms and subsequently use these techniques. As a consequence, these existing works do not provide clarity on the actual performance of clustering methods which may vary depending on the choice of graph matching algorithm used in the first step. This paper studied the clustering problem exactly as encountered in practice.

- Theoretical guarantees are provided for the proposed clustering algorithms and testing framework

Weaknesses:
- The Bernoulli exchangeable model for graph only allows dense graphs (as the probability of edges does not depend on the number of nodes n). Most real world networks are sparse and thus I find this model restrictive.
- Assumption of strict degree monotonicity is a restrictive one which is not satisfied in many real data examples. It appears that other existing histogram methods for graphon estimation such as by Airoldi, Costa and Chan, 2013 or Olhede and Wolfe, 2014 might be better suited for the clustering task given this practical concern.
- Number of clusters K is assumed to be known for the numerical experiments whereas in practice this must be suitably determined from the data itself
-Simulation study does not report performance for the case of sparse networks
- Not clear how robust the techniques are to the assumption on strict monotonicity of the degree not being satisfied and to the choice of metric used to define the graph distance
- A discussion on the consequences of Assumptions 2 and 3 specifically wrt clustering (the main objective of the paper) would have been helpful

Other minor comments:
- From reading the introduction it was not clear what 'clustering' meant exactly : in general for a  population of graphs, it could mean clustering of nodes in each graph or clustering across graphs or both.
 .- What sort of graphs are considered? Directed/self-loops? Not mentioned in the paper.
- What exactly is Assumption 3? Is it assuming that each w_i is an element of an equivalence class unrelated to the equivalence class of w_j? How is the equivalence class defined?

**Summary Of The Paper:**

Motivated by the sort-and-smooth graphon estimator of Chan and Airoldi, 2014, this paper proposes two new clustering algorithms (graph distance based spectral clustering and similarity based semidefinite programming) for multiple graphs observed without vertex correspondence. The idea is to use the graphon approximation method to obtain a histogram estimator with the same number of bins for each graph and subsequently apply an existing spectral clustering approach on the graphon based distance matrix.
Theoretical properties of their approach are studied (under reasonable smoothness assumptions on the generating process i.e. graphon) and consistency is established. The graph distance metric is applied to test for similarity of two collections of graphs.


**Summary Of The Review:**

Overall, the proposed idea combines existing approaches on graphon estimation and spectral clustering to lead to the proposed clustering and testing approach. It is a neat idea and comes with theoretical guarantees, however, I find the setting of dense networks and the strict assumptions on degree sequence as required for graphon estimation, restrictive. The performance of the clustering algorithms and the testing approach when these assumptions fail and when the number of clusters is unknown, is not clear from the paper.

---

> ### Author Response · Authors · 2021-11-14
> **Response to reviewer's concerns (sparse graphs, clarification on Assumptions 2 and 3)**
>
> We thank the reviewer for the positive and constructive feedback. We clarify the concerns raised in the following. Minor comments are directly addressed in the paper.
> ### 1. Graphon does not allow sparse graphs
> We thank the reviewer for pointing this out. We explored the possibility of generating sparse graphs from graphons using the approach from Olhede and Wolfe,2014 and analysed the consistency of the proposed distance. We have included a short discussion under ‘Remark on Proposition 1 for sparse graphs’ in the paper and will include the proof in the appendix shortly. We briefly detail the analysis below.
> * We modify the link probability of the edge between any two nodes $i$ and $j$ of graph $G$ to $G_{ij}|U_i,U_j \sim \mathit{Bernoulli} (\rho w(U_i, U_j))$ where $\rho$ is the expected fraction of edges in $G$, and depends on $n$.
> * Under this definition, we derive the consistency of empirical degree sorting as detailed in Chan and Airoldi,2014 and we find that the consistency result (Lemma 2 in Chan and Airoldi,2014) will hold for the choice of $\rho \geq \sqrt{\dfrac{\log n}{n}}$. The only change in the lemma will be $\left| \dfrac{\sigma(i)}{n} - \dfrac{\sigma(j)}{n} \right| < \dfrac{1}{\rho} \dfrac{1}{6L_1} \sqrt{\dfrac{\log n}{n}}$ and the probability will become $1-8\exp{\left( -\dfrac{1}{18L_1^2} \dfrac{\log n}{\rho^2} \right)}$ in equation 19. Corresponding change is reflected in the converse also.
> * The modified derivation suggests that the algorithms are consistent for $\rho \geq \sqrt{\dfrac{\log n}{n}}$. We acknowledge that the derived bound for $\rho$ is not close to the practical sparsity where $\rho$ should be $\dfrac{\log n}{n}$.
> However, the algorithms performed exceedingly well on real graphs which are sparse. This is also the reason why we did not evaluate our algorithms on simulated sparse graphs.
>
> ### 2. Clarification on Assumptions 2 and 3
> **Assumption 2:**
> We agree that Assumption 2 is strong because of the strictness (degree monotonicity can be obtained by reordering naturally) and we thank the reviewer for bringing up this point.
> We considered different ways (like the ones cited by the reviewer) to develop methods for clustering graphs without this assumption, but the main concern is the practicality of such methods.
> For instance, given two (continuous or different sized) graphons, one can reduce the graphons to $p \times p$ matrices by identifying a fixed number of block structures and then smoothing by averaging [Olhede and Wolfe 2014, Airoldi et al., 2013, Klopp et al., 2017]. Subsequently, we can apply an alignment technique like graph matching on the reduced graphons, followed by analysis similar to the one presented in the paper. Note that the alignment of two graphons approximated to 2 Stochastic Block Model will need $2^{p^2}$ computation.
> Although this would lead to a statistically better result, there will be a tradeoff with practicality because of the computationally inefficient alignment step (also refer to the accuracy and computation performance of graph matching based approach NCGMM).
> Moreover, statistical consistency results for graph matching techniques are rather limited.
>
> On the above grounds, we chose a restrictive setup to facilitate theoretical analysis while providing computational and performance advantage in practice.
> In the context of clustering, the theoretical guarantees on DSC and SSDP (Theorem 1,2) hinge upon the proposed graph distance consistency which require degree monotonicity. Therefore, it is quintessential for the theory, however empirical results showed that the algorithms are efficient even on real datasets where the assumptions are violated, thereby demonstrating that the assumptions (all three) are inconsequential for practical purposes. We briefly mention the above approach in the paper, and leave the question of its efficiency/consistency as an open problem
>
> **Assumption 3**:
> An equivalence class refers to all the graphons resulting in the same random graph model. For example, let us consider graphons $w(u,v) = uv$ and $w^\prime(u,v)=(1-u)(1-v)$. Although $w(u,v)$ and $w^\prime(u,v)$ are functionally different, the random graph models from both the graphons are the same. In the context of clustering, it is important to *not* cluster the graphs from these graphons separately.  This is captured effectively in Assumption 3 as it states that graphons $w_i$ and $w_j$ will result in the same random graph models if and only if $w_i = w_j$. Moreover, it is a mild assumption in the purview of the theoretical results.
> * Olhede and Wolfe. Network histograms and universality of blockmodel approximation. PNAS,2014
> * Chan, Costa, Chan. Stochastic blockmodel approximation of a graphon: Theory and consistent estimation. NIPS,2013
> * Klopp, Tsybakov, Verzelen, et al. Oracle inequalities for network models and sparse graphon estimation. Annals of Statistics,2017
> * Chan and Airoldi. A consistent histogram estimator for exchangeable graph models. ICML,2014

---

> > ### Author Response · Authors · 2021-11-14
> > **Response to reviewer's comments (effect of different metric in graph distance, known K)**
> >
> > ### 3. Effect of different metric in graph distance
> > Our result is tightly coupled to the Frobenius norm of the graph distance and achieving the same result for other metrics would not be possible, since it depends on the consistency of empirical degree sorting. To understand the effect of other metrics, we need to study its impact on the degree sorting.
> >
> > ### 4. Number of clusters $K$ assumed to be known
> > We refer the reviewer to the heading ‘Remark on the knowledge of $K$’ at the end of Section 3.2 where we have addressed this point.

---

### Author Response · Authors · 2021-11-14
**Update in the paper other than reviewer's comments/concerns**

We would like to inform that we have updated the two-sample testing experiments in our paper, in addition to the reviewer's comments and concerns. We compared the proposed test statistics $T$ defined using our graph distance with two other possible distances - log moments from Mukherjee et al.,2017 and MMD, an efficient test statistics for random dot product graphs [Agterberg et al.,2020].
We evaluated the different tests on both simulated and real datasets, and the results can be found in Section 4 and Appendix C

* Soumendu Sundar Mukherjee, Purnamrita Sarkar, and Lizhen Lin. On clustering network-valued data. In Advances in neural information processing systems, pp. 7071–7081, 2017.
* Joshua Agterberg, Minh Tang, and Carey Priebe. Nonparametric two-sample hypothesis testing for random graphs with negative and repeated eigenvalues. arXiv preprint arXiv:2012.09828, 2020.

------------

We have also included the possibility of generating sparse graphs from graphons using the technique mentioned by Reviewer fKkm. The  discussion is added under 'Remark on Proposition 1 for sparse graphs' in the main paper and the proof can be found in Appendix A.2.

---

### Decision · Program_Chairs · 2022-01-20

**Decision:**

Accept (Poster)

**Comment:**

This paper presents a new method for clustering multiple graphs, without vertex correspondence, by combing existing approaches on graphon estimation and spectral clustering. All reviewers agree that this is a neat paper with new theoretical and empirical results. The main concerns were also properly addressed during rebutal. Overall, it is a good paper.